# A-Mem: Agentic Memory for LLM Agents

**Wujiang Xu**[1], **Zujie Liang**[2], **Kai Mei**[1], **Hang Gao**[1], **Juntao Tan**[1], **Yongfeng Zhang**[1,3]

[1]Rutgers University   [2]Independent Researcher   [3]AIOS Foundation

wujiang.xu@rutgers.edu

## Abstract

While large language model (LLM) agents can effectively use external tools for complex real-world tasks, they require memory systems to leverage historical experiences. Current memory systems enable basic storage and retrieval but lack sophisticated memory organization, despite recent attempts to incorporate graph databases. Moreover, these systems' fixed operations and structures limit their adaptability across diverse tasks. To address this limitation, this paper proposes a novel agentic memory system for LLM agents that can dynamically organize memories in an agentic way. Following the basic principles of the Zettelkasten method, we designed our memory system to create interconnected knowledge networks through dynamic indexing and linking. When a new memory is added, we generate a comprehensive note containing multiple structured attributes, including contextual descriptions, keywords, and tags. The system then analyzes historical memories to identify relevant connections, establishing links where meaningful similarities exist. Additionally, this process enables memory evolution – as new memories are integrated, they can trigger updates to the contextual representations and attributes of existing historical memories, allowing the memory network to continuously refine its understanding. Our approach combines the structured organization principles of Zettelkasten with the flexibility of agent-driven decision making, allowing for more adaptive and context-aware memory management. Empirical experiments on six foundation models show superior improvement against existing SOTA baselines.

 **Code for Benchmark Evaluation**:
https://github.com/WujiangXu/AgenticMemory

 **Code for Production-ready Agentic Memory**:
https://github.com/WujiangXu/A-mem-sys

## 1 Introduction

Large Language Model (LLM) agents have demonstrated remarkable capabilities in various tasks, with recent advances enabling them to interact with environments, execute tasks, and make decisions autonomously [23, 33, 7]. They integrate LLMs with external tools and delicate workflows to improve reasoning and planning abilities. Though LLM agent has strong reasoning performance, it still needs a memory system to provide long-term interaction ability with the external environment [35].

Existing memory systems [25, 39, 28, 21] for LLM agents provide basic memory storage functionality. These systems require agent developers to predefine memory storage structures, specify storage points within the workflow, and establish retrieval timing. Meanwhile, to improve structured memory organization, Mem0 [8], following the principles of RAG [9, 18, 30], incorporates graph databases for storage and retrieval processes. While graph databases provide structured organization for memory systems, their reliance on predefined schemas and relationships fundamentally limits their adaptability. This limitation manifests clearly in practical scenarios - when an agent learns a novel mathematical

39th Conference on Neural Information Processing Systems (NeurIPS 2025).

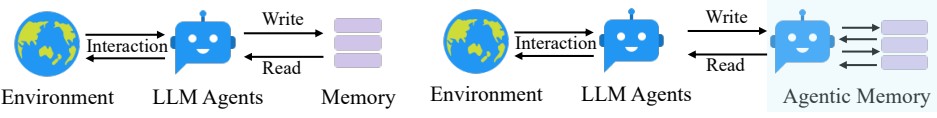

**(a)** Traditional memory system.  **(b)** Our proposed agentic memory.

**Figure 1:** Traditional memory systems require predefined memory access patterns specified in the workflow, limiting their adaptability to diverse scenarios. Contrastly, our A-MEM enhances the flexibility of LLM agents by enabling dynamic memory operations.

solution, current systems can only categorize and link this information within their preset framework, unable to forge innovative connections or develop new organizational patterns as knowledge evolves. Such rigid structures, coupled with fixed agent workflows, severely restrict these systems' ability to generalize across new environments and maintain effectiveness in long-term interactions. The challenge becomes increasingly critical as LLM agents tackle more complex, open-ended tasks, where flexible knowledge organization and continuous adaptation are essential. Therefore, *how to design a flexible and universal memory system that supports LLM agents' long-term interactions* remains a crucial challenge.

In this paper, we introduce a novel agentic memory system, named as A-MEM, for LLM agents that enables dynamic memory structuring without relying on static, predetermined memory operations. Our approach draws inspiration from the Zettelkasten method [15, 1], a sophisticated knowledge management system that creates interconnected information networks through atomic notes and flexible linking mechanisms. Our system introduces an agentic memory architecture that enables autonomous and flexible memory management for LLM agents. For each new memory, we construct comprehensive notes, which integrates multiple representations: structured textual attributes including several attributes and embedding vectors for similarity matching. Then A-MEM analyzes the historical memory repository to establish meaningful connections based on semantic similarities and shared attributes. This integration process not only creates new links but also enables dynamic evolution when new memories are incorporated, they can trigger updates to the contextual representations of existing memories, allowing the entire memories to continuously refine and deepen its understanding over time. The contributions are summarized as:

• We present A-MEM, an agentic memory system for LLM agents that enables autonomous generation of contextual descriptions, dynamic establishment of memory connections, and intelligent evolution of existing memories based on new experiences. This system equips LLM agents with long-term interaction capabilities without requiring predetermined memory operations.

• We design an agentic memory update mechanism where new memories automatically trigger two key operations: link generation and memory evolution. Link generation automatically establishes connections between memories by identifying shared attributes and similar contextual descriptions. Memory evolution enables existing memories to dynamically adapt as new experiences are analyzed, leading to the emergence of higher-order patterns and attributes.

• We conduct comprehensive evaluations of our system using a long-term conversational dataset, comparing performance across six foundation models using six distinct evaluation metrics, demonstrating significant improvements. Moreover, we provide T-SNE visualizations to illustrate the structured organization of our agentic memory system.

## 2 Related Work

### 2.1 Memory for LLM Agents

Prior works on LLM agent memory systems have explored various mechanisms for memory management and utilization [23, 21, 8, 39]. Some approaches complete interaction storage, which maintains comprehensive historical records through dense retrieval models [39] or read-write memory structures [24]. Moreover, MemGPT [25] leverages cache-like architectures to prioritize recent information. Similarly, SCM [32] proposes a Self-Controlled Memory framework that enhances LLMs' capability to maintain long-term memory through a memory stream and controller mechanism. However, these approaches face significant limitations in handling diverse real-world tasks. While they can provide basic memory functionality, their operations are typically constrained by predefined

structures and fixed workflows. These constraints stem from their reliance on rigid operational patterns, particularly in memory writing and retrieval processes. Such inflexibility leads to poor generalization in new environments and limited effectiveness in long-term interactions. Therefore, designing a flexible and universal memory system that supports agents' long-term interactions remains a crucial challenge.

## 2.2 Retrieval-Augmented Generation

Retrieval-Augmented Generation (RAG) has emerged as a powerful approach to enhance LLMs by incorporating external knowledge sources [18, 6, 10]. The standard RAG [37, 34] process involves indexing documents into chunks, retrieving relevant chunks based on semantic similarity, and augmenting the LLM's prompt with this retrieved context for generation. Advanced RAG systems [20, 12] have evolved to include sophisticated pre-retrieval and post-retrieval optimizations. Building upon these foundations, recent researches has introduced agentic RAG systems that demonstrate more autonomous and adaptive behaviors in the retrieval process. These systems can dynamically determine when and what to retrieve [4, 14], generate hypothetical responses to guide retrieval, and iteratively refine their search strategies based on intermediate results [31, 29].

However, while agentic RAG approaches demonstrate agency in the retrieval phase by autonomously deciding when and what to retrieve [4, 14, 38], our agentic memory system exhibits agency at a more fundamental level through the autonomous evolution of its memory structure. Inspired by the Zettelkasten method, our system allows memories to actively generate their own contextual descriptions, form meaningful connections with related memories, and evolve both their content and relationships as new experiences emerge. This fundamental distinction in agency between retrieval versus storage and evolution distinguishes our approach from agentic RAG systems, which maintain static knowledge bases despite their sophisticated retrieval mechanisms.

## 3 Methodolodgy

Our proposed agentic memory system draws inspiration from the Zettelkasten method, implementing a dynamic and self-evolving memory system that enables LLM agents to maintain long-term memory without predetermined operations. The system's design emphasizes atomic note-taking, flexible linking mechanisms, and continuous evolution of knowledge structures.

### 3.1 Note Construction

Building upon the Zettelkasten method's principles of atomic note-taking and flexible organization, we introduce an LLM-driven approach to memory note construction. When an agent interacts with its environment, we construct structured memory notes that capture both explicit information and LLM-generated contextual understanding. Each memory note $m_i$ in our collection $\mathcal{M} = \{m_1, m_2, ..., m_N\}$ is represented as:

$$m_i = \{c_i, t_i, K_i, G_i, X_i, e_i, L_i\} \tag{1}$$

where $c_i$ represents the original interaction content, $t_i$ is the timestamp of the interaction, $K_i$ denotes LLM-generated keywords that capture key concepts, $G_i$ contains LLM-generated tags for categorization, $X_i$ represents the LLM-generated contextual description that provides rich semantic understanding, and $L_i$ maintains the set of linked memories that share semantic relationships. To enrich each memory note with meaningful context beyond its basic content and timestamp, we leverage an LLM to analyze the interaction and generate these semantic components. The note construction process involves prompting the LLM with carefully designed templates $P_{s1}$:

$$K_i, G_i, X_i \leftarrow \text{LLM}(c_i \,\|\, t_i \,\|\, P_{s1}) \tag{2}$$

Following the Zettelkasten principle of atomicity, each note captures a single, self-contained unit of knowledge. To enable efficient retrieval and linking, we compute a dense vector representation via a text encoder [27] that encapsulates all textual components of the note:

$$e_i = f_{\text{enc}}\big[\, \text{concat}(c_i, K_i, G_i, X_i) \,\big] \tag{3}$$

By using LLMs to generate enriched components, we enable autonomous extraction of implicit knowledge from raw interactions. The multi-faceted note structure ($K_i$, $G_i$, $X_i$) creates rich representations that capture different aspects of the memory, facilitating nuanced organization and

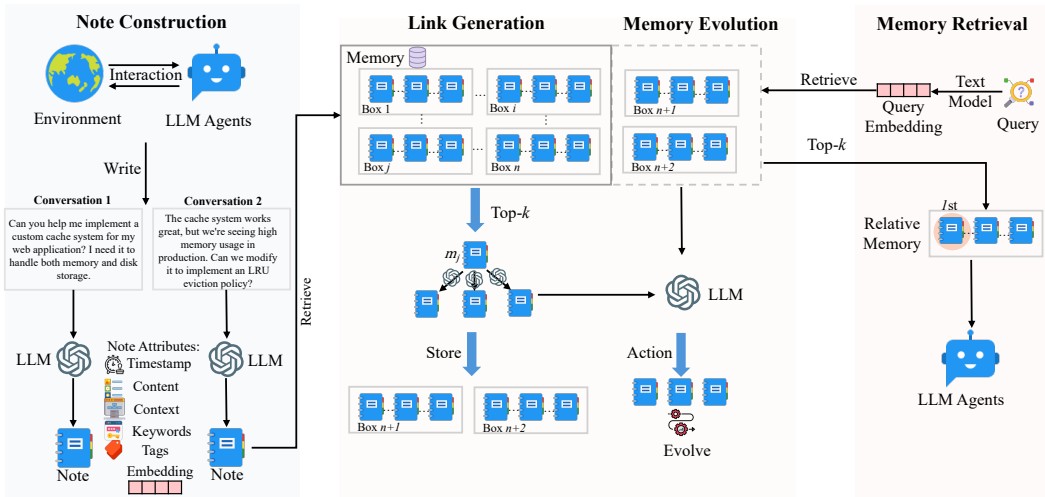

**Figure 2:** Our A-MEM architecture comprises three integral parts in memory storage. During note construction, the system processes new interaction memories and stores them as notes with multiple attributes. The link generation process first retrieves the most relevant historical memories and then employs an LLM to determine whether connections should be established between them. The concept of a 'box' describes that related memories become interconnected through their similar contextual descriptions, analogous to the Zettelkasten method. However, our approach allows individual memories to exist simultaneously within multiple different boxes. During the memory retrieval stage, we extract query embeddings using a text encoding model and search the memory database for relevant matches. When related memory is retrieved, similar memories that are linked within the same box are also automatically accessed.

retrieval. Additionally, the combination of LLM-generated semantic components with dense vector representations provides both context and computationally efficient similarity matching.

## 3.2 Link Generation

Our system implements an autonomous link generation mechanism that enables new memory notes to form meaningful connections without predefined rules. When the constrctd memory note $m_n$ is added to the system, we first leverage its semantic embedding for similarity-based retrieval. For each existing memory note $m_j \in \mathcal{M}$, we compute a similarity score:

$$s_{n,j} = \frac{e_n \cdot e_j}{|e_n||e_j|} \tag{4}$$

The system then identifies the top-$k$ most relevant memories:

$$\mathcal{M}_{\text{near}}^n = \{m_j | \text{rank}(s_{n,j}) \le k, m_j \in \mathcal{M}\} \tag{5}$$

Based on these candidate nearest memories, we prompt the LLM to analyze potential connections based on their potential common attributes. Formally, the link set of memory $m_n$ update like:

$$L_i \leftarrow \text{LLM}(m_n \, \| \mathcal{M}_{\text{near}}^n \, \| P_{s2}) \tag{6}$$

Each generated link $l_i$ is structured as: $L_i = \{m_i, ..., m_k\}$. By using embedding-based retrieval as an initial filter, we enable efficient scalability while maintaining semantic relevance. A-MEM can quickly identify potential connections even in large memory collections without exhaustive comparison. More importantly, the LLM-driven analysis allows for nuanced understanding of relationships that goes beyond simple similarity metrics. The language model can identify subtle patterns, causal relationships, and conceptual connections that might not be apparent from embedding similarity alone. We implements the Zettelkasten principle of flexible linking while leveraging modern language models. The resulting network emerges organically from memory content and context, enabling natural knowledge organization.

## 3.3 Memory Evolution

After creating links for the new memory, A-MEM evolves the retrieved memories based on their textual information and relationships with the new memory. For each memory $m_j$ in the nearest

neighbor set $\mathcal{M}_{\text{near}}^n$, the system determines whether to update its context, keywords, and tags. This evolution process can be formally expressed as:

$$m_j^* \leftarrow \text{LLM}(m_n \,\|\, \mathcal{M}_{\text{near}}^n \setminus m_j \,\|\, m_j \,\|\, P_{s3}) \tag{7}$$

The evolved memory $m_j^*$ then replaces the original memory $m_j$ in the memory set $\mathcal{M}$. This evolutionary approach enables continuous updates and new connections, mimicking human learning processes. As the system processes more memories over time, it develops increasingly sophisticated knowledge structures, discovering higher-order patterns and concepts across multiple memories. This creates a foundation for autonomous memory learning where knowledge organization becomes progressively richer through the ongoing interaction between new experiences and existing memories.

### 3.4 Retrieve Relative Memory

In each interaction, our A-MEM performs context-aware memory retrieval to provide the agent with relevant historical information. Given a query text $q$ from the current interaction, we first compute its dense vector representation using the same text encoder used for memory notes:

$$e_q = f_{\text{enc}}(q) \tag{8}$$

The system then computes similarity scores between the query embedding and all existing memory notes in $\mathcal{M}$ using cosine similarity:

$$s_{q,i} = \frac{e_q \cdot e_i}{|e_q||e_i|}, \text{where } e_i \in m_i, \ \forall m_i \in \mathcal{M} \tag{9}$$

Then we retrieve the k most relevant memories from the historical memory storage to construct a contextually appropriate prompt.

$$\mathcal{M}_{\text{retrieved}} = \{m_i | \text{rank}(s_{q,i}) \le k, m_i \in \mathcal{M}\} \tag{10}$$

These retrieved memories provide relevant historical context that helps the agent better understand and respond to the current interaction. The retrieved context enriches the agent's reasoning process by connecting the current interaction with related past experiences stored in the memory system.

## 4 Experiment

### 4.1 Dataset and Evaluation

To evaluate the effectiveness of instruction-aware recommendation in long-term conversations, we utilize the LoCoMo dataset [22], which contains significantly longer dialogues compared to existing conversational datasets [36, 13]. While previous datasets contain dialogues with around 1K tokens over 4-5 sessions, LoCoMo features much longer conversations averaging 9K tokens spanning up to 35 sessions, making it particularly suitable for evaluating models' ability to handle long-range dependencies and maintain consistency over extended conversations. The LoCoMo dataset comprises diverse question types designed to comprehensively evaluate different aspects of model understanding: (1) single-hop questions answerable from a single session; (2) multi-hop questions requiring information synthesis across sessions; (3) temporal reasoning questions testing understanding of time-related information; (4) open-domain knowledge questions requiring integration of conversation context with external knowledge; and (5) adversarial questions assessing models' ability to identify unanswerable queries. In total, LoCoMo contains 7,512 question-answer pairs across these categories. Besides, we use a new dataset, named DialSim [16], to evaluate the effectiveness of our memory system. It is question-answering dataset derived from long-term multi-party dialogues. The dataset is derived from popular TV shows (Friends, The Big Bang Theory, and The Office), covering 1,300 sessions spanning five years, containing approximately 350,000 tokens, and including more than 1,000 questions per session from refined fan quiz website questions and complex questions generated from temporal knowledge graphs.

For comparison baselines, we compare to **LoCoMo** [22], **ReadAgent** [17], **MemoryBank** [39] and **MemGPT** [25]. The detailed introduction of baselines can be found in Appendix A.1 For evaluation, we employ two primary metrics: the F1 score to assess answer accuracy by balancing precision and recall, and BLEU-1 [26] to evaluate generated response quality by measuring word overlap

**Table 1:** Experimental results on LoCoMo dataset of QA tasks across five categories (Multi Hop, Temporal, Open Domain, Single Hop, and Adversarial) using different methods. Results are reported in F1 and BLEU-1 (%) scores. The best performance is marked in bold, and our proposed method A-MEM (highlighted in gray) demonstrates competitive performance across six foundation language models.

| Model | | Method | Multi Hop | | Temporal | | Open Domain | | Single Hop | | Adversarial | | Ranking | | Token Length |
|---|---|---|---|---|---|---|---|---|---|---|---|---|---|---|---|
| | | | F1 | BLEU | F1 | BLEU | F1 | BLEU | F1 | BLEU | F1 | BLEU | F1 | BLEU | |
| GPT | 4o-mini | LoCoMo | 25.02 | 19.75 | 18.41 | 14.77 | 12.04 | 11.16 | 40.36 | 29.05 | 69.23 | 68.75 | 2.4 | 2.4 | 16,910 |
| | | READAGENT | 9.15 | 6.48 | 12.60 | 8.87 | 5.31 | 5.12 | 9.67 | 7.66 | 9.81 | 9.02 | 4.2 | 4.2 | 643 |
| | | MEMORYBANK | 5.00 | 4.77 | 9.68 | 6.99 | 5.56 | 5.94 | 6.61 | 5.16 | 7.36 | 6.48 | 4.8 | 4.8 | 432 |
| | | MEMGPT | 26.65 | 17.72 | 25.52 | 19.44 | 9.15 | 7.44 | 41.04 | 34.34 | 43.29 | 42.73 | 2.4 | 2.4 | 16,977 |
| | | **A-MEM** | **27.02** | **20.09** | **45.85** | **36.67** | **12.14** | **12.00** | **44.65** | **37.06** | 50.03 | 49.47 | **1.2** | **1.2** | 2,520 |
| | 4o | LoCoMo | 28.00 | 18.47 | 9.09 | 5.78 | 16.47 | 14.80 | 61.56 | 54.19 | 52.61 | 51.13 | 2.0 | 2.0 | 16,910 |
| | | READAGENT | 14.61 | 9.95 | 4.16 | 3.19 | 8.84 | 8.37 | 12.46 | 10.29 | 6.81 | 6.13 | 4.0 | 4.0 | 805 |
| | | MEMORYBANK | 6.49 | 4.69 | 2.47 | 2.43 | 6.43 | 5.30 | 8.28 | 7.10 | 4.42 | 3.67 | 5.0 | 5.0 | 569 |
| | | MEMGPT | 30.36 | 22.83 | 17.29 | 13.18 | 12.24 | 11.87 | 60.16 | 53.35 | 34.96 | 34.25 | 2.4 | 2.4 | 16,987 |
| | | **A-MEM** | **32.86** | **23.76** | **39.41** | **31.23** | **17.10** | **15.84** | 48.43 | 42.97 | 36.35 | 35.53 | **1.6** | **1.6** | 1,216 |
| Qwen2.5 | 1.5b | LoCoMo | 9.05 | 6.55 | 4.25 | 4.04 | 9.91 | 8.50 | 11.15 | 8.67 | 40.38 | 40.23 | 3.4 | 3.4 | 16,910 |
| | | READAGENT | 6.61 | 4.93 | 2.55 | 2.51 | 5.31 | 12.24 | 10.13 | 7.54 | 5.42 | 27.32 | 4.6 | 4.6 | 752 |
| | | MEMORYBANK | 11.14 | 8.25 | 4.46 | 2.87 | 8.05 | 6.21 | 13.42 | 11.01 | 36.76 | 34.00 | 2.6 | 2.6 | 284 |
| | | MEMGPT | 10.44 | 7.61 | 4.21 | 3.89 | 11.14 | 11.64 | 9.56 | 7.34 | 31.51 | 28.90 | 3.4 | 3.4 | 16,953 |
| | | **A-MEM** | **18.23** | **11.94** | **24.32** | **19.74** | **16.48** | **14.31** | **23.63** | **19.23** | **46.00** | **43.26** | **1.0** | **1.0** | 1,300 |
| | 3b | LoCoMo | 4.61 | 4.29 | 3.11 | 2.71 | 4.55 | 5.97 | 7.03 | 5.69 | 16.95 | 14.81 | 3.2 | 3.2 | 16,910 |
| | | READAGENT | 2.47 | 1.78 | 3.01 | 3.01 | 5.57 | 5.22 | 3.25 | 2.51 | 15.78 | 14.01 | 4.2 | 4.2 | 776 |
| | | MEMORYBANK | 3.60 | 3.39 | 1.72 | 1.97 | 6.63 | 6.58 | 4.11 | 3.32 | 13.07 | 10.30 | 4.2 | 4.2 | 298 |
| | | MEMGPT | 5.07 | 4.31 | 2.94 | 2.95 | 7.04 | 7.10 | 7.26 | 5.52 | 14.47 | 12.39 | 2.4 | 2.4 | 16,961 |
| | | **A-MEM** | **12.57** | **9.01** | **27.59** | **25.07** | **7.12** | **7.28** | **17.23** | **13.12** | **27.91** | **25.15** | **1.0** | **1.0** | 1,137 |
| Llama 3.2 | 1b | LoCoMo | 11.25 | 9.18 | 7.38 | 6.82 | 11.90 | 10.38 | 12.86 | 10.50 | 51.89 | 48.27 | 3.4 | 3.4 | 16,910 |
| | | READAGENT | 5.96 | 5.12 | 1.93 | 2.30 | 12.46 | 11.17 | 7.75 | 6.03 | 44.64 | 40.15 | 4.6 | 4.6 | 665 |
| | | MEMORYBANK | 13.18 | 10.03 | 7.61 | 6.27 | 15.78 | 12.94 | 17.30 | 14.03 | 52.61 | 47.53 | 2.0 | 2.0 | 274 |
| | | MEMGPT | 9.19 | 6.96 | 4.02 | 4.79 | 11.14 | 8.24 | 10.16 | 7.68 | 49.75 | 45.11 | 4.0 | 4.0 | 16,950 |
| | | **A-MEM** | **19.06** | **11.71** | **17.80** | **10.28** | **17.55** | **14.67** | **28.51** | **24.13** | **58.81** | **54.28** | **1.0** | **1.0** | 1,376 |
| | 3b | LoCoMo | 6.88 | 5.77 | 4.37 | 4.40 | 10.65 | 9.29 | 8.37 | 6.93 | 30.25 | 28.46 | 2.8 | 2.8 | 16,910 |
| | | READAGENT | 2.47 | 1.78 | 3.01 | 3.01 | 5.57 | 5.22 | 3.25 | 2.51 | 15.78 | 14.01 | 4.2 | 4.2 | 461 |
| | | MEMORYBANK | 6.19 | 4.47 | 3.49 | 3.13 | 4.07 | 4.57 | 7.61 | 6.03 | 18.65 | 17.05 | 3.2 | 3.2 | 263 |
| | | MEMGPT | 5.32 | 3.99 | 2.68 | 2.72 | 5.64 | 5.54 | 4.32 | 3.51 | 21.45 | 19.37 | 3.8 | 3.8 | 16,956 |
| | | **A-MEM** | **17.44** | **11.74** | **26.38** | **19.50** | **12.53** | **11.83** | **28.14** | **23.87** | **42.04** | **40.60** | **1.0** | **1.0** | 1,126 |

with ground truth responses. Also, we report the average token length for answering one question. Besides reporting experiment results with four additional metrics (ROUGE-L, ROUGE-2, METEOR, and SBERT Similarity), we also present experimental outcomes using different foundation models including DeepSeek-R1-32B [11], Claude 3.0 Haiku [2], and Claude 3.5 Haiku [3] in Appendix A.3.

## 4.2 Implementation Details

For all baselines and our proposed method, we maintain consistency by employing identical system prompts as detailed in Appendix B. The deployment of Qwen-1.5B/3B and Llama 3.2 1B/3B models is accomplished through local instantiation using Ollama [1], with LiteLLM [2] managing structured output generation. For GPT models, we utilize the official structured output API. In our memory retrieval process, we primarily employ $k=10$ for top-$k$ memory selection to maintain computational efficiency, while adjusting this parameter for specific categories to optimize performance. The detailed configurations of $k$ can be found in Appendix A.5. For text embedding, we implement the all-minilm-l6-v2 model across all experiments.

## 4.3 Emprical Results

**Performance Analysis.** In our empirical evaluation, we compared A-MEM with four competitive baselines including LoCoMo [22], ReadAgent [17], MemoryBank [39], and MemGPT [25] on the LoCoMo dataset. For non-GPT foundation models, our A-MEM consistently outperforms all baselines across different categories, demonstrating the effectiveness of our agentic memory approach. For GPT-based models, while LoCoMo and MemGPT show strong performance in certain categories like Open Domain and Adversarial tasks due to their robust pre-trained knowledge in simple fact retrieval, our A-MEM demonstrates superior performance in Multi-Hop tasks achieves at least two times better performance that require complex reasoning chains. In addition to experiments on the LoCoMo dataset, we also compare our method on the DialSim dataset against LoCoMo and MemGPT. A-MEM consistently outperforms all baselines across evaluation metrics, achieving an F1

---

[1] https://github.com/ollama/ollama
[2] https://github.com/BerriAI/litellm

**Table 2:** Comparison of different memory mechanisms across multiple evaluation metrics on DialSim [16]. Higher scores indicate better performance, with A-MEM showing superior results across all metrics.

| Method | F1 | BLEU-1 | ROUGE-L | ROUGE-2 | METEOR | SBERT Similarity |
|---|---|---|---|---|---|---|
| LoCoMo | 2.55 | 3.13 | 2.75 | 0.90 | 1.64 | 15.76 |
| MemGPT | 1.18 | 1.07 | 0.96 | 0.42 | 0.95 | 8.54 |
| **A-MEM** | **3.45** | **3.37** | **3.54** | **3.60** | **2.05** | **19.51** |

**Table 3:** An ablation study was conducted to evaluate our proposed method against the GPT-4o-mini base model. The notation 'w/o' indicates experiments where specific modules were removed. The abbreviations LG and ME denote the link generation module and memory evolution module, respectively.

| Method | Category | | | | | | | | | |
|---|---|---|---|---|---|---|---|---|---|---|
| | Multi Hop | | Temporal | | Open Domain | | Single Hop | | Adversial | |
| | F1 | BLEU-1 | F1 | BLEU-1 | F1 | BLEU-1 | F1 | BLEU-1 | F1 | BLEU-1 |
| w/o LG & ME | 9.65 | 7.09 | 24.55 | 19.48 | 7.77 | 6.70 | 13.28 | 10.30 | 15.32 | 18.02 |
| w/o ME | 21.35 | 15.13 | 31.24 | 27.31 | 10.13 | 10.85 | 39.17 | 34.70 | 44.16 | 45.33 |
| **A-MEM** | **27.02** | **20.09** | **45.85** | **36.67** | **12.14** | **12.00** | **44.65** | **37.06** | **50.03** | **49.47** |

score of 3.45 (a 35% improvement over LoCoMo's 2.55 and 192% higher than MemGPT's 1.18). The effectiveness of A-MEM stems from its novel agentic memory architecture that enables dynamic and structured memory management. Unlike traditional approaches that use static memory operations, our system creates interconnected memory networks through atomic notes with rich contextual descriptions, enabling more effective multi-hop reasoning. The system's ability to dynamically establish connections between memories based on shared attributes and continuously update existing memory descriptions with new contextual information allows it to better capture and utilize the relationships between different pieces of information.

**Cost-Efficiency Analysis.** A-MEM demonstrates significant computational and cost efficiency alongside strong performance. The system requires approximately 1,200 tokens per memory operation, achieving an 85-93% reduction in token usage compared to baseline methods (LoCoMo and MemGPT with 16,900 tokens) through our selective top-k retrieval mechanism. This substantial token reduction directly translates to lower operational costs, with each memory operation costing less than $0.0003 when using commercial API services—making large-scale deployments economically viable. Processing times average 5.4 seconds using GPT-4o-mini and only 1.1 seconds with locally-hosted Llama 3.2 1B on a single GPU. Despite requiring multiple LLM calls during memory processing, A-MEM maintains this cost-effective resource utilization while consistently outperforming baseline approaches across all foundation models tested, particularly doubling performance on complex multi-hop reasoning tasks. This balance of low computational cost and superior reasoning capability highlights A-MEM's practical advantage for deployment in the real world.

## 4.4 Ablation Study

To evaluate the effectiveness of the Link Generation (LG) and Memory Evolution (ME) modules, we conduct the ablation study by systematically removing key components of our model. When both LG and ME modules are removed, the system exhibits substantial performance degradation, particularly in Multi Hop reasoning and Open Domain tasks. The system with only LG active (w/o ME) shows intermediate performance levels, maintaining significantly better results than the version without both modules, which demonstrates the fundamental importance of link generation in establishing memory connections. Our full model, A-MEM, consistently achieves the best performance across all evaluation categories, with particularly strong results in complex reasoning tasks. These results reveal that while the link generation module serves as a critical foundation for memory organization, the memory evolution module provides essential refinements to the memory structure. The ablation study validates our architectural design choices and highlights the complementary nature of these two modules in creating an effective memory system.

## 4.5 Hyperparameter Analysis

We conducted extensive experiments to analyze the impact of the memory retrieval parameter k, which controls the number of relevant memories retrieved for each interaction. As shown in Figure 3, we evaluated performance across different k values (10, 20, 30, 40, 50) on five categories of tasks using GPT-4o-mini as our base model. The results reveal an interesting pattern: while increasing k generally leads to improved performance, this improvement gradually plateaus and sometimes slightly decreases at higher values. This trend is particularly evident in Multi Hop and Open Domain

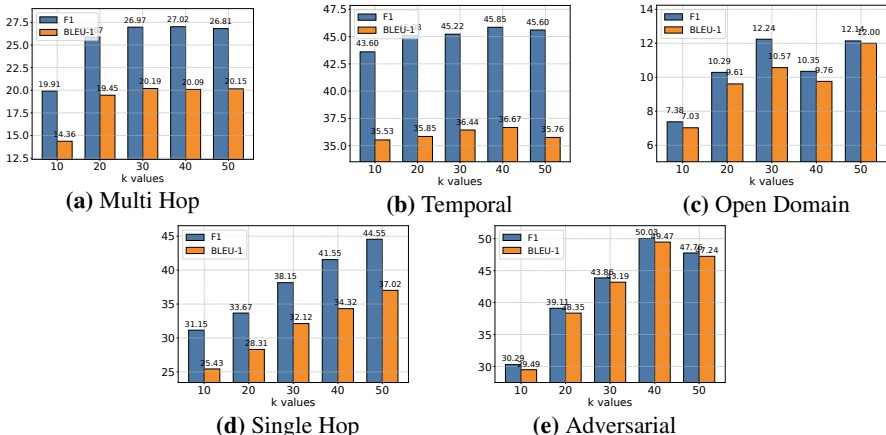

**Figure 3:** Impact of memory retrieval parameter k across different task categories with GPT-4o-mini as the base model. While larger k values generally improve performance by providing richer historical context, the gains diminish beyond certain thresholds, suggesting a trade-off between context richness and effective information processing. This pattern is consistent across all evaluation categories, indicating the importance of balanced context retrieval for optimal performance.

**Table 4:** Comparison of memory usage and retrieval time across different memory methods and scales.

| Memory Size | Method | Memory Usage (MB) | Retrieval Time ($\mu$s) |
|---|---|---|---|
| 1,000 | A-MEM | 1.46 | 0.31 ± 0.30 |
| | MemoryBank [39] | 1.46 | 0.24 ± 0.20 |
| | ReadAgent [17] | 1.46 | 43.62 ± 8.47 |
| 10,000 | A-MEM | 14.65 | 0.38 ± 0.25 |
| | MemoryBank [39] | 14.65 | 0.26 ± 0.13 |
| | ReadAgent [17] | 14.65 | 484.45 ± 93.86 |
| 100,000 | A-MEM | 146.48 | 1.40 ± 0.49 |
| | MemoryBank [39] | 146.48 | 0.78 ± 0.26 |
| | ReadAgent [17] | 146.48 | 6,682.22 ± 111.63 |
| 1,000,000 | A-MEM | 1464.84 | 3.70 ± 0.74 |
| | MemoryBank [39] | 1464.84 | 1.91 ± 0.31 |
| | ReadAgent [17] | 1464.84 | 120,069.68 ± 1,673.39 |

tasks. The observation suggests a delicate balance in memory retrieval - while larger k values provide richer historical context for reasoning, they may also introduce noise and challenge the model's capacity to process longer sequences effectively. Our analysis indicates that moderate k values strike an optimal balance between context richness and information processing efficiency.

## 4.6 Scaling Analysis

To evaluate storage costs with accumulating memory, we examined the relationship between storage size and retrieval time across our A-MEM system and two baseline approaches: MemoryBank [39] and ReadAgent [17]. We evaluated these three memory systems with identical memory content across four scale points, increasing the number of entries by a factor of 10 at each step (from 1,000 to 10,000, 100,000, and finally 1,000,000 entries). The experimental results reveal key insights about our A-MEM system's scaling properties: In terms of space complexity, all three systems exhibit identical linear memory usage scaling ($O(N)$), as expected for vector-based retrieval systems. This confirms that A-MEM introduces no additional storage overhead compared to baseline approaches. For retrieval time, A-MEM demonstrates excellent efficiency with minimal increases as memory size grows. Even when scaling to 1 million memories, A-MEM's retrieval time increases only from $0.31\mu$s to $3.70\mu$s, representing exceptional performance. While MemoryBank shows slightly faster retrieval times, A-MEM maintains comparable performance while providing richer memory representations and functionality. Based on our space complexity and retrieval time analysis, we conclude that A-MEM's retrieval mechanisms maintain excellent efficiency even at large scales. The minimal growth in retrieval time across memory sizes addresses concerns about efficiency in large-scale memory systems, demonstrating that A-MEM provides a highly scalable solution for long-term conversation management. This unique combination of efficiency, scalability, and enhanced memory capabilities positions A-MEM as a significant advancement in building powerful and long-term memory mechanism for LLM Agents.

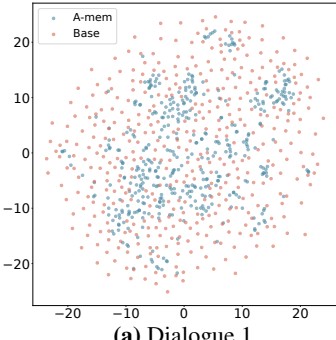 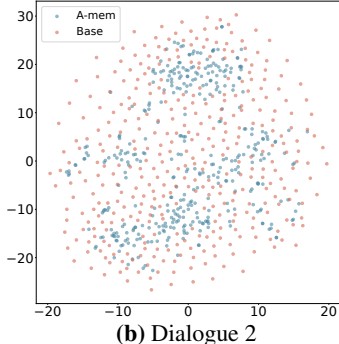

**(a)** Dialogue 1                **(b)** Dialogue 2

**Figure 4:** T-SNE Visualization of Memory Embeddings Showing More Organized Distribution with A-MEM (blue) Compared to Base Memory (red) Across Different Dialogues. Base Memory represents A-MEM without link generation and memory evolution.

## 4.7 Memory Analysis

We present the t-SNE visualization in Figure 4 of memory embeddings to demonstrate the structural advantages of our agentic memory system. Analyzing two dialogues sampled from long-term conversations in LoCoMo [22], we observe that A-MEM (shown in blue) consistently exhibits more coherent clustering patterns compared to the baseline system (shown in red). This structural organization is particularly evident in Dialogue 2, where well-defined clusters emerge in the central region, providing empirical evidence for the effectiveness of our memory evolution mechanism and contextual description generation. In contrast, the baseline memory embeddings display a more dispersed distribution, demonstrating that memories lack structural organization without our link generation and memory evolution components. These visualization results validate that A-MEM can autonomously maintain meaningful memory structures through dynamic evolution and linking mechanisms. More results can be seen in Appendix A.4.

## 5 Conclusions

In this work, we introduced A-MEM, a novel agentic memory system that enables LLM agents to dynamically organize and evolve their memories without relying on predefined structures. Drawing inspiration from the Zettelkasten method, our system creates an interconnected knowledge network through dynamic indexing and linking mechanisms that adapt to diverse real-world tasks. The system's core architecture features autonomous generation of contextual descriptions for new memories and intelligent establishment of connections with existing memories based on shared attributes. Furthermore, our approach enables continuous evolution of historical memories by incorporating new experiences and developing higher-order attributes through ongoing interactions. Through extensive empirical evaluation across six foundation models, we demonstrated that A-MEM achieves superior performance compared to existing state-of-the-art baselines in long-term conversational tasks. Visualization analysis further validates the effectiveness of our memory organization approach. These results suggest that agentic memory systems can significantly enhance LLM agents' ability to utilize long-term knowledge in complex environments.

## 6 Limitations

While our agentic memory system achieves promising results, we acknowledge several areas for potential future exploration. First, although our system dynamically organizes memories, the quality of these organizations may still be influenced by the inherent capabilities of the underlying language models. Different LLMs might generate slightly different contextual descriptions or establish varying connections between memories. Additionally, while our current implementation focuses on text-based interactions, future work could explore extending the system to handle multimodal information, such as images or audio, which could provide richer contextual representations.

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

# Contents

# APPENDIX

## A  Experiment

### A.1  Detailed Baselines Introduction

**LoCoMo** [22] takes a direct approach by leveraging foundation models without memory mechanisms for question answering tasks. For each query, it incorporates the complete preceding conversation and questions into the prompt, evaluating the model's reasoning capabilities.

**ReadAgent** [17] tackles long-context document processing through a sophisticated three-step methodology: it begins with episode pagination to segment content into manageable chunks, followed by memory gisting to distill each page into concise memory representations, and concludes with interactive look-up to retrieve pertinent information as needed.

**MemoryBank** [39] introduces an innovative memory management system that maintains and efficiently retrieves historical interactions. The system features a dynamic memory updating mechanism based on the Ebbinghaus Forgetting Curve theory, which intelligently adjusts memory strength according to time and significance. Additionally, it incorporates a user portrait building system that progressively refines its understanding of user personality through continuous interaction analysis.

**MemGPT** [25] presents a novel virtual context management system drawing inspiration from traditional operating systems' memory hierarchies. The architecture implements a dual-tier structure: a main context (analogous to RAM) that provides immediate access during LLM inference, and an external context (analogous to disk storage) that maintains information beyond the fixed context window.

### A.2  Evaluation Metric

The F1 score represents the harmonic mean of precision and recall, offering a balanced metric that combines both measures into a single value. This metric is particularly valuable when we need to balance between complete and accurate responses:

$$\text{F1} = 2 \cdot \frac{\text{precision} \cdot \text{recall}}{\text{precision} + \text{recall}} \tag{11}$$

where

$$\text{precision} = \frac{\text{true positives}}{\text{true positives} + \text{false positives}} \tag{12}$$

$$\text{recall} = \frac{\text{true positives}}{\text{true positives} + \text{false negatives}} \tag{13}$$

In question-answering systems, the F1 score serves a crucial role in evaluating exact matches between predicted and reference answers. This is especially important for span-based QA tasks, where systems must identify precise text segments while maintaining comprehensive coverage of the answer.

BLEU-1 [26] provides a method for evaluating the precision of unigram matches between system outputs and reference texts:

$$\text{BLEU-1} = BP \cdot \exp\left( \sum_{n=1}^{1} w_n \log p_n \right) \tag{14}$$

where

$$BP = \begin{cases} 1 & \text{if } c > r \\ e^{1-r/c} & \text{if } c \leq r \end{cases} \tag{15}$$

$$p_n = \frac{\sum_i \sum_k \min(h_{ik}, m_{ik})}{\sum_i \sum_k h_{ik}} \tag{16}$$

Here, $c$ is candidate length, $r$ is reference length, $h_{ik}$ is the count of n-gram i in candidate k, and $m_{ik}$ is the maximum count in any reference. In QA, BLEU-1 evaluates the lexical precision of generated answers, particularly useful for generative QA systems where exact matching might be too strict.

ROUGE-L [19] measures the longest common subsequence between the generated and reference texts.

$$\text{ROUGE-L} = \frac{(1 + \beta^2)R_l P_l}{R_l + \beta^2 P_l} \tag{17}$$

$$R_l = \frac{\text{LCS}(X, Y)}{|X|} \tag{18}$$

$$P_l = \frac{\text{LCS}(X, Y)}{|Y|} \tag{19}$$

where $X$ is reference text, $Y$ is candidate text, and LCS is the Longest Common Subsequence.

ROUGE-2 [19] calculates the overlap of bigrams between the generated and reference texts.

$$\text{ROUGE-2} = \frac{\sum_{\text{bigram} \in \text{ref}} \min(\text{Count}_{\text{ref}}(\text{bigram}), \text{Count}_{\text{cand}}(\text{bigram}))}{\sum_{\text{bigram} \in \text{ref}} \text{Count}_{\text{ref}}(\text{bigram})} \tag{20}$$

Both ROUGE-L and ROUGE-2 are particularly useful for evaluating the fluency and coherence of generated answers, with ROUGE-L focusing on sequence matching and ROUGE-2 on local word order.

METEOR [5] computes a score based on aligned unigrams between the candidate and reference texts, considering synonyms and paraphrases.

$$\text{METEOR} = F_{\text{mean}} \cdot (1 - \text{Penalty}) \tag{21}$$

$$F_{\text{mean}} = \frac{10P \cdot R}{R + 9P} \tag{22}$$

$$\text{Penalty} = 0.5 \cdot \left(\frac{\text{ch}}{m}\right)^3 \tag{23}$$

where $P$ is precision, $R$ is recall, ch is number of chunks, and $m$ is number of matched unigrams. METEOR is valuable for QA evaluation as it considers semantic similarity beyond exact matching, making it suitable for evaluating paraphrased answers.

SBERT Similarity [27] measures the semantic similarity between two texts using sentence embeddings.

$$\text{SBERT\_Similarity} = \cos(\text{SBERT}(x), \text{SBERT}(y)) \tag{24}$$

$$\cos(a, b) = \frac{a \cdot b}{\|a\|\|b\|} \tag{25}$$

SBERT($x$) represents the sentence embedding of text. SBERT Similarity is particularly useful for evaluating semantic understanding in QA systems, as it can capture meaning similarities even when the lexical overlap is low.

### A.3 Comparison Results

Our comprehensive evaluation using ROUGE-2, ROUGE-L, METEOR, and SBERT metrics demonstrates that A-MEM achieves superior performance while maintaining remarkable computational efficiency. Through extensive empirical testing across various model sizes and task categories, we have established A-MEM as a more effective approach compared to existing baselines, supported by several compelling findings. In our analysis of non-GPT models, specifically Qwen2.5 and Llama 3.2, A-MEM consistently outperforms all baseline approaches across all metrics. The Multi-Hop category showcases particularly striking results, where Qwen2.5-15b with A-MEM achieves a ROUGE-L score of 27.23, dramatically surpassing LoComo's 4.68 and ReadAgent's 2.81 - representing a nearly six-fold improvement. This pattern of superiority extends consistently across METEOR and SBERT

Table 5: Experimental results on LoCoMo dataset of QA tasks across five categories (Multi Hop, Temporal, Open Domain, Single Hop, and Adversial) using different methods. Results are reported in ROUGE-2 and ROUGE-L scores, abbreviated to RGE-2 and RGE-L. The best performance is marked in bold, and our proposed method A-Mem (highlighted in gray) demonstrates competitive performance across six foundation language models.

| Model | | Method | Category | | | | | | | | | |
|---|---|---|---|---|---|---|---|---|---|---|---|---|
| | | | Multi Hop | | Temporal | | Open Domain | | Single Hop | | Adversial | |
| | | | RGE-2 | RGE-L | RGE-2 | RGE-L | RGE-2 | RGE-L | RGE-2 | RGE-L | RGE-2 | RGE-L |
| GPT | 4o-mini | LoCoMo | 9.64 | 23.92 | 2.01 | 18.09 | 3.40 | 11.58 | 26.48 | 40.20 | **60.46** | **69.59** |
| | | ReadAgent | 2.47 | 9.45 | 0.95 | 13.12 | 0.55 | 5.76 | 2.99 | 9.92 | 6.66 | 9.79 |
| | | MemoryBank | 1.18 | 5.43 | 0.52 | 9.64 | 0.97 | 5.77 | 1.64 | 6.63 | 4.55 | 7.35 |
| | | MemGPT | 10.58 | 25.60 | 4.76 | 25.22 | 0.76 | 9.14 | 28.44 | 42.24 | 36.62 | 43.75 |
| | | **A-Mem** | **10.61** | **25.86** | **21.39** | **44.27** | **3.42** | **12.09** | **29.50** | **45.18** | 42.62 | 50.04 |
| | 4o | LoCoMo | 11.53 | 30.65 | 1.68 | 8.17 | 3.21 | 16.33 | **45.42** | **63.86** | **45.13** | **52.67** |
| | | ReadAgent | 3.91 | 14.36 | 0.43 | 3.96 | 0.52 | 8.58 | 4.75 | 13.41 | 4.24 | 6.81 |
| | | MemoryBank | 1.84 | 7.36 | 0.36 | 2.29 | 2.13 | 6.85 | 3.02 | 9.35 | 1.22 | 4.41 |
| | | MemGPT | 11.55 | 30.18 | 4.66 | 15.83 | 3.27 | 14.02 | 43.27 | 62.75 | 28.72 | 35.08 |
| | | **A-Mem** | **12.76** | **31.71** | **9.82** | **25.04** | **6.09** | **16.63** | 33.67 | 50.31 | 30.31 | 36.34 |
| Qwen2.5 | 1.5b | LoCoMo | 1.39 | 9.24 | 0.00 | 4.68 | 3.42 | 10.59 | 3.25 | 11.15 | 35.10 | 43.61 |
| | | ReadAgent | 0.74 | 7.14 | 0.10 | 2.81 | 3.05 | 12.63 | 1.47 | 7.88 | 20.73 | 27.82 |
| | | MemoryBank | 1.51 | 11.18 | 0.14 | 5.39 | 1.80 | 8.44 | 5.07 | 13.72 | 29.24 | 36.95 |
| | | MemGPT | 1.16 | 11.35 | 0.00 | 7.88 | 2.87 | 14.62 | 2.18 | 9.82 | 23.96 | 31.69 |
| | | **A-Mem** | **4.88** | **17.94** | **5.88** | **27.23** | **3.44** | **16.87** | **12.32** | **24.38** | **36.32** | **46.60** |
| | 3b | LoCoMo | 0.49 | 4.83 | 0.14 | 3.20 | 1.31 | 5.38 | 1.97 | 6.98 | 12.66 | 17.10 |
| | | ReadAgent | 0.08 | 4.08 | 0.00 | 1.96 | 1.26 | 6.19 | 0.73 | 4.34 | 7.35 | 10.64 |
| | | MemoryBank | 0.43 | 3.76 | 0.05 | 1.61 | 0.24 | 6.32 | 1.03 | 4.22 | 9.55 | 13.41 |
| | | MemGPT | 0.69 | 5.55 | 0.05 | 3.17 | 1.90 | 7.90 | 2.05 | 7.32 | 10.46 | 14.39 |
| | | **A-Mem** | **2.91** | **12.42** | **8.11** | **27.74** | 1.51 | 7.51 | **8.80** | **17.57** | **21.39** | **27.98** |
| Llama 3.2 | 1b | LoCoMo | 2.51 | 11.48 | 0.44 | 8.25 | 1.69 | 13.06 | 2.94 | 13.00 | 39.85 | 52.74 |
| | | ReadAgent | 0.53 | 6.49 | 0.00 | 4.62 | 5.47 | 14.29 | 1.19 | 8.03 | 34.52 | 45.55 |
| | | MemoryBank | 2.96 | 13.57 | 0.23 | 10.53 | 4.01 | 18.38 | 6.41 | 17.66 | 41.15 | 53.31 |
| | | MemGPT | 1.82 | 9.91 | 0.06 | 6.56 | 2.13 | 11.36 | 2.00 | 10.37 | 38.59 | 50.31 |
| | | **A-Mem** | **4.82** | **19.31** | **1.84** | **20.47** | **5.99** | **18.49** | **14.82** | **29.78** | **46.76** | **60.23** |
| | 3b | LoCoMo | 0.98 | 7.22 | 0.03 | 4.45 | 2.36 | 11.39 | 2.85 | 8.45 | 25.47 | 30.26 |
| | | ReadAgent | 2.47 | 1.78 | 3.01 | 3.01 | 5.07 | 5.22 | 3.25 | 2.51 | 15.78 | 14.01 |
| | | MemoryBank | 1.83 | 6.96 | 0.25 | 3.41 | 0.43 | 4.43 | 2.73 | 7.83 | 14.64 | 18.59 |
| | | MemGPT | 0.72 | 5.39 | 0.11 | 2.85 | 0.61 | 5.74 | 1.45 | 4.42 | 16.62 | 21.47 |
| | | **A-Mem** | **6.02** | **17.62** | **7.93** | **27.97** | **5.38** | **13.00** | **16.89** | **28.55** | **35.48** | **42.25** |

scores. When examining GPT-based models, our results reveal an interesting pattern. While LoComo and MemGPT demonstrate strong capabilities in Open Domain and Adversarial tasks, A-Mem shows remarkable superiority in Multi-Hop reasoning tasks. Using GPT-4o-mini, A-Mem achieves a ROUGE-L score of 44.27 in Multi-Hop tasks, more than doubling LoComo's 18.09. This significant advantage maintains consistency across other metrics, with METEOR scores of 23.43 versus 7.61 and SBERT scores of 70.49 versus 52.30. The significance of these results is amplified by A-Mem's exceptional computational efficiency. Our approach requires only 1,200-2,500 tokens, compared to the substantial 16,900 tokens needed by LoComo and MemGPT. This efficiency stems from two key architectural innovations: First, our novel agentic memory architecture creates interconnected memory networks through atomic notes with rich contextual descriptions, enabling more effective capture and utilization of information relationships. Second, our selective top-k retrieval mechanism facilitates dynamic memory evolution and structured organization. The effectiveness of these innovations is particularly evident in complex reasoning tasks, as demonstrated by the consistently strong Multi-Hop performance across all evaluation metrics. Besides, we also show the experimental results with different foundational models including DeepSeek-R1-32B [11], Claude 3.0 Haiku [2] and Claude 3.5 Haiku [3].

## A.4 Memory Analysis

In addition to the memory visualizations of the first two dialogues shown in the main text, we present additional visualizations in Fig.5 that demonstrate the structural advantages of our agentic memory system. Through analysis of two dialogues sampled from long-term conversations in LoCoMo[22], we observe that A-Mem (shown in blue) consistently produces more coherent clustering patterns compared to the baseline system (shown in red). This structural organization is particularly evident in Dialogue 2, where distinct clusters emerge in the central region, providing empirical support for the effectiveness of our memory evolution mechanism and contextual description generation. In contrast, the baseline memory embeddings exhibit a more scattered distribution, indicating that memories lack structural organization without our link generation and memory evolution components.

**Table 6:** Experimental results on LoCoMo dataset of QA tasks across five categories (Multi Hop, Temporal, Open Domain, Single Hop, and Adversial) using different methods. Results are reported in METEOR and SBERT Similarity scores, abbreviated to ME and SBERT. The best performance is marked in bold, and our proposed method A-MEM (highlighted in gray) demonstrates competitive performance across six foundation language models.

| Model | | Method | Multi Hop | | Temporal | | Open Domain | | Single Hop | | Adversial | |
|---|---|---|---|---|---|---|---|---|---|---|---|---|
| | | | ME | SBERT | ME | SBERT | ME | SBERT | ME | SBERT | ME | SBERT |
| GPT | 4o-mini | LoCoMo | 15.81 | 47.97 | 7.61 | 52.30 | 8.16 | 35.00 | 40.42 | 57.78 | **63.28** | **71.93** |
| | | ReadAgent | 5.46 | 28.67 | 4.76 | 45.07 | 3.69 | 26.72 | 8.01 | 26.78 | 8.38 | 15.20 |
| | | MemoryBank | 3.42 | 21.71 | 4.07 | 37.58 | 4.21 | 23.71 | 5.81 | 20.76 | 6.24 | 13.00 |
| | | MemGPT | 15.79 | 49.33 | 13.25 | 61.53 | 4.59 | 32.77 | 41.40 | 58.19 | 39.16 | 47.24 |
| | | A-Mem | **16.36** | **49.46** | **23.43** | **70.49** | **8.36** | **38.48** | **42.32** | **59.38** | 45.64 | 53.26 |
| | 4o | LoCoMo | 16.34 | 53.82 | 7.21 | 32.15 | 8.98 | **43.72** | **53.39** | **73.40** | **47.72** | **56.09** |
| | | ReadAgent | 7.86 | 37.41 | 3.76 | 26.22 | 4.42 | 30.75 | 9.36 | 31.37 | 5.47 | 12.34 |
| | | MemoryBank | 3.22 | 26.23 | 2.29 | 23.49 | 4.18 | 24.89 | 6.64 | 23.90 | 2.93 | 10.01 |
| | | MemGPT | 16.64 | 55.12 | 12.68 | 35.93 | 7.78 | 37.91 | 52.14 | 72.83 | 31.15 | 39.08 |
| | | A-Mem | **17.53** | **55.96** | **13.10** | **45.40** | **10.62** | 38.87 | 41.93 | 62.47 | 32.34 | 40.11 |
| Qwen2.5 | 1.5b | LoCoMo | 4.99 | 32.23 | 2.86 | 34.03 | 5.89 | 35.61 | 8.57 | 29.47 | 40.53 | 50.49 |
| | | ReadAgent | 3.67 | 28.20 | 1.88 | 27.27 | 8.97 | 35.13 | 5.52 | 26.33 | 24.04 | 34.12 |
| | | MemoryBank | 5.57 | 35.40 | 2.80 | 32.47 | 4.27 | 33.85 | 10.59 | 32.16 | 32.93 | 42.83 |
| | | MemGPT | 5.40 | 35.64 | 2.35 | 39.04 | 7.68 | 40.36 | 7.07 | 30.16 | 27.24 | 40.63 |
| | | A-Mem | **9.49** | **43.49** | **11.92** | **61.65** | **9.11** | **42.58** | **19.69** | **41.93** | **40.64** | **52.44** |
| | 3b | LoCoMo | 2.00 | 24.37 | 1.92 | 25.24 | 3.45 | 25.38 | 6.00 | 21.28 | 16.67 | 23.14 |
| | | ReadAgent | 1.78 | 21.10 | 1.69 | 20.78 | 4.43 | 25.15 | 3.37 | 18.20 | 10.46 | 17.39 |
| | | MemoryBank | 2.37 | 17.81 | 2.22 | 21.93 | 3.86 | 20.65 | 3.99 | 16.26 | 15.49 | 20.77 |
| | | MemGPT | 3.74 | 24.31 | 2.25 | 27.67 | 6.44 | 29.59 | 6.24 | 22.40 | 13.19 | 20.83 |
| | | A-Mem | **6.25** | **33.72** | **14.04** | **62.54** | **6.56** | **30.60** | **15.98** | **33.98** | **27.36** | **33.72** |
| Llama 3.2 | 1b | LoCoMo | 5.77 | 38.02 | 3.38 | 45.44 | 6.20 | 42.69 | 9.33 | 34.19 | 46.79 | 60.74 |
| | | ReadAgent | 2.97 | 29.26 | 1.31 | 26.45 | 7.13 | 39.19 | 5.36 | 26.44 | 42.39 | 54.35 |
| | | MemoryBank | 6.77 | 39.33 | 4.43 | 45.63 | 7.76 | 42.81 | 13.01 | 37.32 | 50.43 | 60.81 |
| | | MemGPT | 5.10 | 32.99 | 2.54 | 41.81 | 3.26 | 35.99 | 6.62 | 30.68 | 45.00 | 61.33 |
| | | A-Mem | **9.01** | **45.16** | **7.50** | **54.79** | **8.30** | **43.42** | **22.46** | **47.07** | **53.72** | **68.00** |
| | 3b | LoCoMo | 3.69 | 27.94 | 2.96 | 20.40 | 6.46 | 32.17 | 6.58 | 22.92 | 29.02 | 35.74 |
| | | ReadAgent | 1.21 | 17.40 | 2.33 | 12.02 | 3.39 | 19.63 | 2.46 | 14.63 | 14.37 | 21.25 |
| | | MemoryBank | 3.84 | 25.06 | 2.73 | 13.65 | 3.05 | 21.08 | 6.35 | 22.02 | 17.14 | 24.39 |
| | | MemGPT | 2.78 | 22.06 | 2.21 | 14.97 | 3.63 | 23.18 | 3.47 | 17.81 | 20.50 | 26.87 |
| | | A-Mem | **9.74** | **39.32** | **13.19** | **59.70** | **8.09** | **32.27** | **24.30** | **42.86** | **39.74** | **46.76** |

**Table 7:** Experimental results on LoCoMo dataset of QA tasks across five categories (Multi Hop, Temporal, Open Domain, Single Hop, and Adversial) using different methods. Results are reported in F1 and BLEU-1 (%) scores with different foundation models.

| Method | Multi Hop | | Temporal | | Open Domain | | Single Hop | | Adversial | |
|---|---|---|---|---|---|---|---|---|---|---|
| | F1 | BLEU-1 | F1 | BLEU-1 | F1 | BLEU-1 | F1 | BLEU-1 | F1 | BLEU-1 |
| **DeepSeek-R1-32B** | | | | | | | | | | |
| LoCoMo | 8.58 | 6.48 | 4.79 | 4.35 | 12.96 | 12.52 | 10.72 | 8.20 | 21.40 | 20.23 |
| MemGPT | 8.28 | 6.25 | 5.45 | 4.97 | 10.97 | 9.09 | 11.34 | 9.03 | **30.77** | **29.23** |
| A-Mem | **15.02** | **10.64** | **14.64** | **11.01** | **14.81** | **12.82** | **15.37** | **12.30** | 27.92 | 27.19 |
| **Claude 3.0 Haiku** | | | | | | | | | | |
| LoCoMo | 4.56 | 3.33 | 0.82 | 0.59 | 2.86 | 3.22 | 3.56 | 3.24 | 3.46 | 3.42 |
| MemGPT | 7.65 | 6.36 | 1.65 | 1.26 | 7.41 | 6.64 | 8.60 | 7.29 | 7.66 | 7.37 |
| A-Mem | **19.28** | **14.69** | **16.65** | **12.23** | **11.85** | **9.61** | **34.72** | **30.05** | **35.99** | **34.87** |
| **Claude 3.5 Haiku** | | | | | | | | | | |
| LoCoMo | 11.34 | 8.21 | 3.29 | 2.69 | 3.79 | 3.58 | 14.01 | 12.57 | 7.37 | 7.12 |
| MemGPT | 8.27 | 6.55 | 3.99 | 2.76 | 4.71 | 4.48 | 16.52 | 14.89 | 5.64 | 5.45 |
| A-Mem | **29.70** | **23.19** | **31.54** | **27.53** | **11.42** | **9.47** | **42.60** | **37.41** | 13.65 | 12.71 |

These visualizations validate that A-Mem can autonomously maintain meaningful memory structures through its dynamic evolution and linking mechanisms.

## A.5  Hyperparameters setting

All hyperparameter k values are presented in Table 8. For models that have already achieved state-of-the-art (SOTA) performance with k=10, we maintain this value without further tuning.

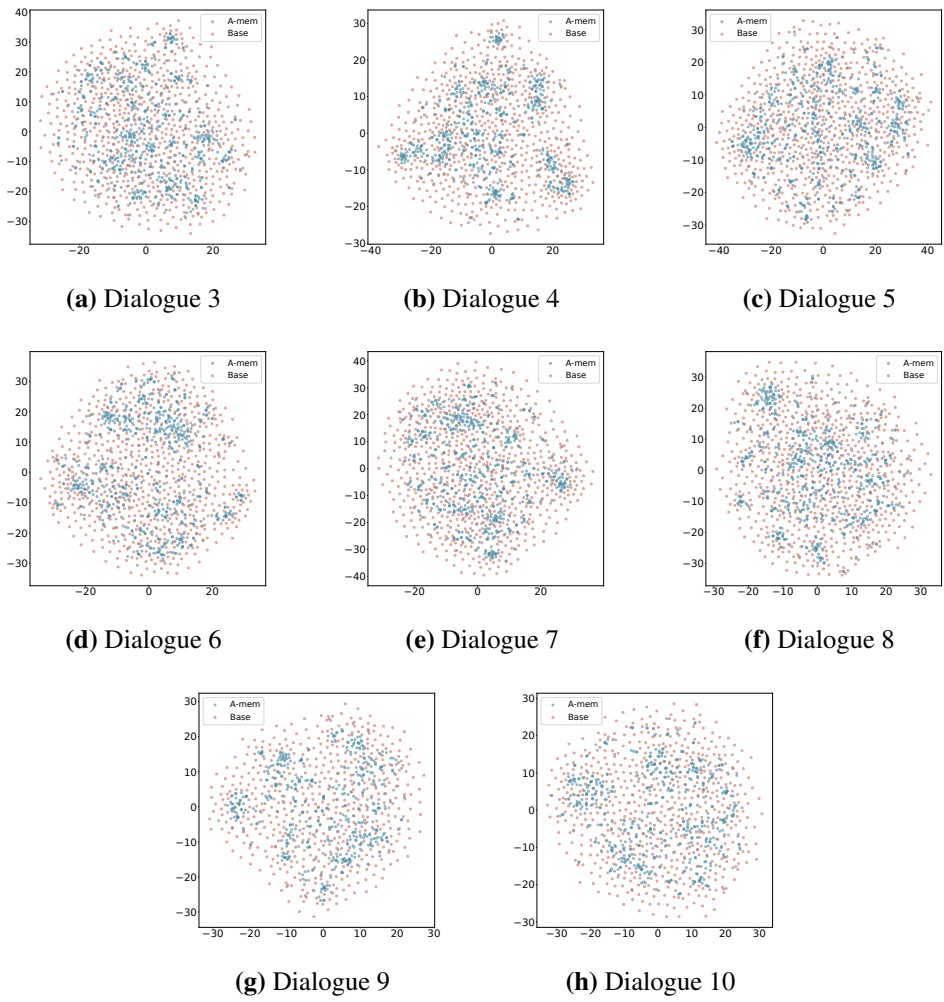

**Figure 5:** T-SNE Visualization of Memory Embeddings Showing More Organized Distribution with A-MEM (blue) Compared to Base Memory (red) Across Different Dialogues. Base Memory represents A-MEM without link generation and memory evolution.

**Table 8:** Selection of k values in retriever across specific categories and model choices.

| Model | Multi Hop | Temporal | Open Domain | Single Hop | Adversial |
|---|---|---|---|---|---|
| GPT-4o-mini | 40 | 40 | 50 | 50 | 40 |
| GPT-4o | 40 | 40 | 50 | 50 | 40 |
| Qwen2.5-1.5b | 10 | 10 | 10 | 10 | 10 |
| Qwen2.5-3b | 10 | 10 | 50 | 10 | 10 |
| Llama3.2-1b | 10 | 10 | 10 | 10 | 10 |
| Llama3.2-3b | 10 | 20 | 10 | 10 | 10 |

# B Prompt Templates and Examples

## B.1 Prompt Template of Note Construction

**The prompt template in Note Construction:** $P_{s1}$
```
Generate a structured analysis of the following content by:
1.  Identifying the most salient keywords (focus on nouns, verbs, and
key concepts)
2.  Extracting core themes and contextual elements
3.  Creating relevant categorical tags
Format the response as a JSON object:
{
"keywords":  [ // several specific, distinct keywords that capture
key concepts and terminology // Order from most to least important //
Don't include keywords that are the name of the speaker or time // At
least three keywords, but don't be too redundant.  ],
"context":  // one sentence summarizing:  // - Main topic/domain // -
Key arguments/points // - Intended audience/purpose ,
"tags":  [ // several broad categories/themes for classification //
Include domain, format, and type tags // At least three tags, but
don't be too redundant.  ]
}
Content for analysis:
```

## B.2 Prompt Template of Link Generation

**The prompt template in Link Generation:** $P_{s2}$
```
You are an AI memory evolution agent responsible for managing and
evolving a knowledge base.
Analyze the the new memory note according to keywords and context,
also with their several nearest neighbors memory.
The new memory context:
{context} content:  {content}
keywords:  {keywords}
The nearest neighbors memories:  {nearest_neighbors_memories}
Based on this information, determine:
Should this memory be evolved?  Consider its relationships with other
memories.
```

## B.3 Prompt Template of Memory Evolution

**The prompt template in Memory Evolution:** $P_{s3}$
```
You are an AI memory evolution agent responsible for managing and
evolving a knowledge base.
Analyze the the new memory note according to keywords and context,
also with their several nearest neighbors memory.
Make decisions about its evolution.
The new memory context:{context}
content:  {content}
keywords:  {keywords}
The nearest neighbors memories:{nearest_neighbors_memories}
Based on this information, determine:
1.  What specific actions should be taken (strengthen,
update_neighbor)?
1.1 If choose to strengthen the connection, which memory should it be
connected to?  Can you give the updated tags of this memory?
1.2 If choose to update neighbor, you can update the context and tags
of these memories based on the understanding of these memories.
Tags should be determined by the content of these characteristic
of these memories, which can be used to retrieve them later and
categorize them.
All the above information should be returned in a list format
according to the sequence:  [[new_memory],[neighbor_memory_1],
...[neighbor_memory_n]]
These actions can be combined.
Return your decision in JSON format with the following structure:  {{
"should_evolve":  true/false,
"actions":  ["strengthen", "merge", "prune"],
"suggested_connections":  ["neighbor_memory_ids"],
"tags_to_update":  ["tag_1",..."tag_n"],
"new_context_neighborhood":  ["new context",...,"new context"],
"new_tags_neighborhood":  [["tag_1",...,"tag_n"],...["tag_1",...,"tag_n"]],
}}
```

## B.4 Examples of Q/A with A-Mem

**Example:**
```
Question 686:  Which hobby did Dave pick up in October 2023?
Prediction:  photography
Reference:  photography
talk start time:10:54 am on 17 November, 2023
memory content:  Speaker Davesays :  Hey Calvin, long time no talk!
A lot has happened.  I've taken up photography and it's been great -
been taking pics of the scenery around here which is really cool.
memory context:  The main topic is the speaker's new hobby of
photography, highlighting their enjoyment of capturing local
scenery, aimed at engaging a friend in conversation about personal
experiences.
memory keywords:  ['photography', 'scenery', 'conversation',
'experience', 'hobby']
memory tags:  ['hobby', 'photography', 'personal development',
'conversation', 'leisure']
talk start time:6:38 pm on 21 July, 2023
memory content:  Speaker Calvinsays :  Thanks, Dave!  It feels
great having my own space to work in.  I've been experimenting
with different genres lately, pushing myself out of my comfort zone.
Adding electronic elements to my songs gives them a fresh vibe.  It's
been an exciting process of self-discovery and growth!
memory context:  The speaker discusses their creative process in
music, highlighting experimentation with genres and the incorporation
of electronic elements for personal growth and artistic evolution.
memory keywords:  ['space', 'experimentation', 'genres', 'electronic',
'self-discovery', 'growth']
memory tags:  ['music', 'creativity', 'self-improvement', 'artistic
expression']
```

