# OpenReview forum: "A-Mem: Agentic Memory for LLM Agents"
_NeurIPS.cc/2025/Conference — NeurIPS 2025 poster_

### Official Review · Reviewer_rgsn · 2025-06-20

**Clarity:** 3
**Significance:** 3
**Originality:** 4
**Rating:** 5
**Confidence:** 3

**Summary:**

This paper proposes A-MEM, an agentic memory system for LLM agents inspired by the Zettelkasten method. It structures each interaction as a contextualized note with semantic attributes and automatically establishes links to relevant past memories. New memories can trigger updates to existing ones, enabling dynamic memory evolution. Experiments on long-term dialogue datasets demonstrate that A-MEM outperforms existing baselines, particularly in multi-hop reasoning tasks, while maintaining high efficiency and scalability.

**Questions:**

1. In the Methodology section, it remains unclear whether the interaction between the environment and the LLM agent is equivalent to a query and the LLM’s response. If these processes are indeed the same, it would be helpful to unify the terminology and revise Figure 2 accordingly to avoid potential confusion for readers.
2. Does the proposed method update the memory after every single interaction? If many interactions do not meaningfully impact the memory, could forcing updates via the LLM lead to degraded performance? What are the advantages of using the LLM for memory evolution compared to alternative approaches, and are there potential negative effects?
3. In the Scaling Analysis, does the reported runtime include the time required for memory evolution? If so, providing a breakdown of the timing for each component would enhance the credibility of the efficiency claims. If not, this omission might render the comparison unfair. It would be beneficial to include an analysis of how the retrieval time scales with the hyperparameter k.

**Ethical Concerns:**

["NO or VERY MINOR ethics concerns only"]

**Final Justification:**

Good paper, after rebuttal, several issues have been addressed.

**Limitations:**

Yes

**Quality:**

3

**Strengths And Weaknesses:**

Strengths:
1. Originality and Significance: The proposed A-MEM method is both novel and innovative, introducing a new memory paradigm characterized by a concise structure and well-motivated design. Compared to existing approaches, it notably supports evolutionary memory capabilities.
2. Quality: The experiments are comprehensive, involving multiple baselines, diverse evaluation metrics, varied dataset types, and different generative models. Furthermore, the authors provide the experiment of computational efficiency, strengthening the practical relevance of the approach.
3. Clarity: The manuscript is well-written and accessible, facilitating reader comprehension.

Weaknesses:
1. Clarity: The description of Methodolodgy Section lacks completeness and clarity in certain aspects. Specific concerns are detailed in the questions 1 and 2.
2. Quality:  For Experiment Section：
1) The Scaling Analysis and Memory Analysis sections are not sufficiently convincing. In particular, the Scaling Analysis suffers from the issues highlighted in question 3. The description of Memory Analysis is somewhat vague, raising doubts about the claim that “A-MEM can autonomously maintain meaningful memory structures.”
2) Apart from the main experiments with LOCOMO, other experiments do not consistently compare against all four baselines, including the primary experiment on the DialSim dataset.
3) Many experiments lack in-depth analysis explaining the reasons behind the model’s superior performance.
3. Significance：The use of a same encoder for structurally different memory and query inputs appears overly simplistic, suggesting that the model design does not adequately account for the heterogeneity of these content.

---

> ### Author Rebuttal · Authors · 2025-07-31
>
> Thank you for your thoughtful and comprehensive evaluation of our work. We are deeply grateful for your recognition of A-Mem's originality and significance. We greatly appreciate your positive assessment of our comprehensive evaluation with multiple baselines, diverse metrics, and computational efficiency analysis. We are also thankful for your recognition of the manuscript's clarity and accessibility.
>
> ***
> > ***Q1. Whether the interaction between the environment and the LLM agent is equivalent to a query and the LLM’s response?***
>
> In Figure 2, the interaction between the environment and LLM agents is not equivalent to a query and the LLM's response. A chatbot, such as GPT and Claude, is one type of LLM agent that has more datasets to evaluate. However, our A-mem can be used in any kind of agent scenarios, such as coding LLM agents, embodied L/VLM agents, etc.
>
> The environment represents the external world or task domain that the agent operates within, providing observations, rewards, and feedback based on the agent's actions. An interaction encompasses the complete cycle of the agent perceiving the environment state, taking an action, and receiving feedback or rewards [1]. Our A-mem serves a dual purpose: it records the entire process of these interactions as contextualized memory notes, and simultaneously provides relevant historical context information during each subsequent interaction to inform better decision-making.
>
> However, due to the limitations of available long-term memory datasets, we primarily evaluate our approach using chatbot agent scenarios in LoCoMo and DialSum. To further demonstrate the effectiveness of our A-mem approach in agent systems, we conduct experiments across two scenarios: QA search and web shopping agents. As shown in the following table, compared to LoCoMo, our A-mem achieves comparable performance while using only 33%-50% of the token budget, demonstrating its ability to provide more informative memory representations. The full experiment details can be found in the response to Reviewer 3BrE Q6.
>
>
> ***
> > ***Q2. Does the proposed method update the memory after every single interaction? Whether forcing updates via the LLM lead to degraded performance in some situations? What are the advantages of using the LLM for memory evolution?***
>
> **Storage Mechanism.** The memory storage strategy can be designed to store and update memory at different frequencies. If LLM agents prefer coarse-grained memories (to save tokens and reduce computational overhead), the saving frequency can be set to a lower rate, such as at the episode level or after multiple sessions, rather than at each individual session.
>
> **Update Control.**  The memory evolution process operates in two stages: first, retrieving the top-k most similar memories based on text embedding similarity, then updating them via LLMs. The retrieval process is governed by a similarity threshold that determines which memories qualify for updates. By adjusting this threshold, we can control the frequency and scope of memory evolution—a higher threshold results in fewer, more selective updates, while a lower threshold allows more frequent updates across a broader range of memories.
>
> In our benchmark evaluations, most memories exhibit strong interconnections, so we did not impose a retrieval threshold. However, for real-world deployment where memory relationships may be more sparse or varied, the retrieval threshold serves as a crucial hyperparameter. By setting an appropriate threshold, A-mem can avoid updating irrelevant or weakly related memories, thereby preventing potentially harmful updates that could degrade memory quality.
>
> **Advantage.** Leveraging LLMs for memory updates allows the existing memory repository to continuously adapt and form associations with new memories, thereby capturing scenario-specific information more accurately. Additionally, the inherent pre-training capabilities of LLMs enable robust memory updating across diverse, general-purpose contexts.
>
>
> ***
> > ***Q3. What is the time for each part of constructing memory? Could you provide an analysis of how the retrieval time scales with the hyperparameter k?***
>
> For link generation and memory evolution, these two operations, the time cost includes two parts including the LLM calls and the retrieval time. For the LLM calls using GPT-4o-mini, the time is an average of 2.4s, which has a strong relation to your api level. For the retrieval time, it can refer to Table 4 in our paper.
>
> Besides, we investigate the retrieval time scales with various k values. The retrieval time remains constant with respect to k due to fundamental algorithmic properties. The retrieval operation consists of four sequential components with distinct complexity characteristics. Query encoding operates at $O(1)$ with respect to k since the query embedding is computed once regardless of k. Similarity computation requires $O(N)$ time with respect to k, as cosine similarity must be computed against the entire corpus of N documents. Global sorting operates at $O(N \log N)$ with respect to k since the similarity scores for all N documents require sorting. Finally, top-k selection requires $O(k)$ time with respect to k for extracting the k highest-scoring results.
>
> The computational bottleneck lies in the first three steps, which are invariant to k. Similarity computation dominates runtime as it requires $O(N)$ dot products between query and document embeddings. Sorting complexity depends on corpus size N, not retrieval size k. Top-k selection contributes negligible overhead, typically less than 0.1% of total time. Therefore, the total retrieval time follows:
>
> $$\text{Total Time} \approx T_{\text{embedding}} + T_{\text{similarity}} + T_{\text{sorting}} + \varepsilon(k)$$
>
> where $\varepsilon(k)$ is asymptotically negligible compared to the N-dependent terms. The key insight is that retrieval systems evaluate the entire corpus regardless of k value, with only the final extraction step scaling with k. This makes k selection performance-neutral, allowing optimization based on task requirements rather than computational constraints, since throughput depends on corpus size N rather than result set size k.
>
>
> ***
> > ***Q4. Other experiments do not consistently compare against all four baselines in DialSim.***
>
> We provide additional experimental results from other baselines on the DialSim dataset. Across six evaluation metrics, we can conclude that our A-mem achieves the best performance.
>
> | Method | F1 | BLEU-1 | ROUGE-L | ROUGE-2 | METEOR | SBERT Similarity |
> |---|---|---|---|---|---|---|
> | ReadAgent | 2.38 | 3.04 | 2.20 | 0.17 | 1.37 | 17.77 |
> | MemoryBank | 1.85 | 3.17 | 1.72 | 0.23 | 1.03 | 9.93 |
> | **A-Mem** | **3.45** | **3.37** | **3.54** | **3.60** | **2.05** | **19.51** |
>
> ***
> > ***Q5. In-depth analysis explaining the reasons behind the model’s superior performance.***
>
> Our A-Mem system demonstrates superior performance by identifying meaningful semantic connections between different memories. Case studies from our memory system reveal how the two-stage mechanism captures nuanced connections that pure embedding approaches miss.
>
> **Case 1.** Our system connects two memories from Caroline's adoption journey. The first shows Caroline reaching out: "Hey Mel, what's up? Long time no see! I just contacted my mentor for adoption advice. I'm ready to be a mom and share my love and family." Nine days later, the second captures her progress: "Woohoo Melanie! I passed the adoption agency interviews last Friday! I'm so excited and thankful." This connection captures a clear temporal progression from preparation to achievement, demonstrating a causal relationship and narrative continuity in her adoption process.
>
> **Case 2.** Our system links Caroline's artistic advocacy evolution. It connects her sharing a painting while explaining "Representing inclusivity and diversity in my art is important to me. I also use it to speak up for the LGBTQ+ community" with her announcement eleven days later: "I'm putting together an LGBTQ art show next month and I'm gonna show my paintings. Super stoked!" This captures her progression from individual artistic expression to community action, showing how personal practice evolves into public advocacy.
>
> These connections emerge because our LLM analysis stage recognizes thematic continuity, temporal coherence, and causal relationships that embedding similarity alone cannot capture, successfully identifying meaningful relationships through contextual understanding rather than lexical overlap.
>
> ***
> > ***Q6. The use of the same encoder for structurally different memory and query inputs appears overly simplistic, which may overlook the heterogeneity of this content.***
>
> Our approach is simple yet effective. Using a unified text embedder provides universal applicability with strong generalization—modern embedders trained on massive heterogeneous datasets demonstrate superior performance across diverse domains without requiring manual feature engineering [2]. Their excellent transfer learning capabilities enable broad applicability across real-world scenarios, providing the flexibility essential for practical deployment [3].
>
> While future work could explore specialized architectures like graph neural networks to capture additional structural information, our current design strikes an optimal balance between strong performance, broad applicability, and ease of deployment.
>
> References:
>
> [1] Sutton, R. S., & Barto, A. G. (2018). The agent-environment interface. In Reinforcement learning: An introduction (Chapter 3.1). MIT press.
>
> [2] Wang, Liang, et al. "Text embeddings by weakly-supervised contrastive pre-training." arXiv preprint arXiv:2212.03533 (2022).
>
> [3] Cer, Daniel, et al. "Universal sentence encoder." arXiv preprint arXiv:1803.11175 (2018).

---

> > ### Comment · Reviewer_rgsn · 2025-08-05
> >
> > Thanks for your detailed response, I will raise my overall score.

---

> > > ### Author Response · Authors · 2025-08-05
> > >
> > > Thank you for taking the time to review our detailed response. We sincerely appreciate your positive feedback and your decision to raise your overall score.

---

### Official Review · Reviewer_3BrE · 2025-06-20

**Clarity:** 3
**Significance:** 3
**Originality:** 3
**Rating:** 4
**Confidence:** 4

**Summary:**

This paper introduces a novel agentic memory system called A-Mem, designed to enhance the performance of Large Language Model (LLM) agents in complex tasks. Drawing principles from the Zettelkasten method, A-Mem overcomes the limitations of existing memory systems, such as fixed structures and adaptability issues, by creating an interconnected knowledge network through dynamic note construction, link generation, and memory evolution. Experiments demonstrate that A-Mem achieves significant improvements in handling long-term conversational tasks compared to existing baselines, particularly excelling in complex tasks like multi-hop reasoning, while also showcasing its cost-effectiveness and scalability.

**Questions:**

1. Is it reasonable about equation (3)? If it comes from Zettlekasten, how about removing 1 or 2 of the 4 factors? Are the results worse?
2. Why does A-Mem perform worse in retrieval time than Memory Bank in Table 4?

**Ethical Concerns:**

["NO or VERY MINOR ethics concerns only"]

**Limitations:**

The potentially biggest issue is that the method proposed in this paper has not been actually applied to existing agents or multi-agent systems (MAS). It's difficult to demonstrate the effectiveness of this method for agents or MAS solely based on the experiments presented in this paper.

**Quality:**

3

**Strengths And Weaknesses:**

Strengths:
1. Novel idea.
2. Dynamic memory organization and strong adaptability.
3. Low resource consumption.

Weaknesses:
1. It does not offer a better memory retrieval strategy, still relying on the most basic RAG method.
2. Establishing links between memories solely based on similarity might not be entirely reasonable.
3. The scaling experiments and hyperparameter experiments do not provide a good impression. The **k** in hyperparameters should arguably be part of scaling, but performance deteriorates when **k** is large; scaling purely on entity number is meaningless, as it simply creates an extremely large document.

---

> ### Author Rebuttal · Authors · 2025-07-31
>
> Thank you for your positive evaluation of our work. We appreciate your recognition of our novel agentic memory system A-Mem, the effectiveness of our Zettelkasten-inspired approach with dynamic note construction, link generation, and memory evolution, and our ability to overcome the limitations of existing memory systems through dynamic memory organization. We are grateful for your acknowledgment of A-Mem's significant improvements in long-term conversational tasks and multi-hop reasoning, its strong adaptability, and its cost-effectiveness with low resource consumption. We address the points you raised to further enhance the clarity of our work.
>
> ***
> > ***Q1. Can A-mem provide a better memory retrieval strategy, rather than the most basic RAG method?***
>
> Unlike simple RAG methods, our A-mem approach employs a distinctive memory retrieval mechanism. As illustrated in Figure 2, during the memory retrieval stage, we extract query embeddings using a text encoding model and search the memory database for relevant matches. When related memory is retrieved, similar memories that are linked
> within the same box are also automatically accessed.
> We will revise Section 3.4 for clarity.
>
> ***
> > ***Q2. Is establishing links between memories solely based on similarity entirely reasonable?***
>
> **Design philosophy.** Our approach uses a two-stage mechanism that balances computational efficiency with semantic understanding. First, embedding similarity identifies top-k candidate memories from the existing collection, providing efficient filtering that scales without exhaustive comparisons. Second, an LLM analyzes these candidates to determine meaningful connections based on contextual relationships and semantic content in an agentic way. This hybrid architecture leverages the computational efficiency of vector similarity for initial retrieval while employing LLM reasoning to detect subtle patterns, causal relationships, and conceptual connections that transcend simple cosine similarity.
>
> **Case study.** Case studies from our memory system reveal how the two-stage mechanism captures nuanced connections that pure embedding approaches miss.
>
> **Case 1.** Our system connects two memories from Caroline's adoption journey. The first shows Caroline reaching out: "Hey Mel, what's up? Long time no see! I just contacted my mentor for adoption advice. I'm ready to be a mom and share my love and family." Nine days later, the second captures her progress: "Woohoo Melanie! I passed the adoption agency interviews last Friday! I'm so excited and thankful." This connection captures a clear temporal progression from preparation to achievement, demonstrating a causal relationship and narrative continuity in her adoption process.
>
> **Case 2.** Our system links Caroline's artistic advocacy evolution. It connects her sharing a painting while explaining, "Representing inclusivity and diversity in my art is important to me. I also use it to speak up for the LGBTQ+ community" with her announcement eleven days later: "I'm putting together an LGBTQ art show next month and I'm gonna show my paintings. Super stoked!" This captures her progression from individual artistic expression to community action, showing how personal practice evolves into public advocacy.
>
> These connections emerge because our LLM analysis stage recognizes thematic continuity, temporal coherence, and causal relationships that embedding similarity alone cannot capture, successfully identifying meaningful relationships through contextual understanding rather than lexical overlap.
>
> ***
> > ***Q3. Why does it not achieve the best performance in scaling experiments compared to memorybank?***
>
> For the scaling analysis, we provide a scalability assessment of our memory system. When we increase the memory size, the retrieval time increases minimally rather than intensely. Although retrieval time is slightly higher than MemoryBank [1], this is because our A-mem stores more contextual information to describe each memory node, including keywords, tags, and context. However, the increased time is quite small per request, while our A-mem provides richer memory representations that yield at least a two-fold improvement in LoCoMo comparison experiments. Based on both performance and scaling analysis, we conclude that our A-mem represents a significant advancement in building powerful and long-term memory mechanisms for LLM agents.
>
> ***
> > ***Q4. Why does increasing the k parameter not lead to improvement?***
>
> In the initial version of the paper, we regard the k value, which controls the number of relevant memories retrieved for each interaction, as a hyperparameter due to its direct influence on the balance between memory coverage and retrieval precision, requiring empirical tuning to achieve optimal performance across different scenarios.
>
> When increasing the value of k, the length of input tokens to the LLM increases correspondingly, which may degrade LLM performance. As discussed in prior research [2-3], large language models suffer from fundamental architectural limitations, including finite context windows, quadratic attention complexity, and the "lost in the middle" problem, where performance degrades significantly when accessing information in the middle of long contexts, even for explicitly long-context models.
>
> In the final version of the paper, we will integrate these two analyses into one section for clarity.
>
> ***
> > ***Q5. Is it reasonable for equation (3)? If it comes from Zettlekasten, how about removing 1 or 2 of the 4 factors? Are the results worse?***
>
> For the vector representation of memory notes, we utilize the attributes of context, content, keywords, and tags, which capture multiple perspectives of each memory note. To validate the effectiveness of this design, we conduct an ablation study by systematically removing each component. Here, $c$ represents the original content, $K$ denotes the keywords, $G$ represents the tags, and $X$ indicates the context. As shown in the table, each component demonstrates its effectiveness in the overall representation, with content and context being particularly impactful.
>
> **Results are reported in F1 and BLEU-1 (%) scores with foundational model GPT-4o-mini.**
>
> | Model | Single Hop |  | Multi Hop |  | Temporal |  | Open Domain |  | Adversial |  |
> |---|---|---|---|---|---|---|---|---|---|---|
> | | F1 | BLEU-1 | F1 | BLEU-1 | F1 | BLEU-1 | F1 | BLEU-1 | F1 | BLEU-1 | F1 | BLEU-1 |
> | -$c$ | 17.20 | 13.10 | 14.84 | 10.93 | 10.73 | 8.57 | 31.54 | 27.06 | 32.08 | 31.34 |
> | -$K$ | 18.76 | 14.36 | 16.15 | 11.98 | 11.56 | 9.37 | 33.68 | 29.35 | 35.04 | 34.07 |
> | -$G$ | 18.76 | 14.28 | 16.22 | 11.88 | 11.51 | 9.36 | 33.83 | 29.28 | 35.03 | 34.02 |
> | -$X$ | 18.12 | 13.74 | 15.48 | 11.35 | 10.97 | 9.01 | 32.37 | 27.94 | 33.43 | 32.41 |
> | **A-MEM** | **19.28** | **14.69** | **16.65** | **12.23** | **11.85** | **9.61** | **34.72** | **30.05** | **35.99** | **34.87** |
>
> ***
> > ***Q6. What are the experimental results in the agent systems?***
>
> To further demonstrate the effectiveness of our A-mem approach in agent systems, we conduct experiments across two scenarios: QA search and web shopping agents.
>
> **Experiments Setup.** For the QA search task, we use a mixture of data from HotpotQA [4] and Natural Questions [5]. We evaluate performance using exact match accuracy and multi-objective F1 scores, comparing against the LoCoMo baseline [6] under two conditions: with and without token truncation. When multiple ground truths are available, we select the maximum F1 score across all alternatives. For multi-objective tasks, the final F1 score represents the sum of F1 scores across all sub-questions. We also report token usage statistics for all methods.
>
> For the web shopping scenario, we evaluate on the WebShop [7] benchmark, measuring performance using the reward scores provided by the environment simulator.
>
> **Results Analysis.** As shown in the following table, compared to LoCoMo, our A-mem achieves comparable performance while using only 33%-50% of the token budget, demonstrating its ability to provide more informative memory representations. This efficiency stems from our memory system's capacity to dynamically establish connections between memories based on shared attributes and continuously update existing memory descriptions with new contextual information, enabling better capture and utilization of inter-informational relationships.
>
> **Results on the QA agent task.**
>
> | Method | Foundation Model |EM |F1 | Token ($*10^2$) |
> |---|---|---|---|---|
> | LoCoMo (Truncate) |Qwen2.5-7B  | 0.396 | 0.497 | **13.3** |
> | LoCoMo |Qwen2.5-7B | 0.165 | 0.213 | 43.3 |
> | LoCoMo |Qwen2.5-14B | 0.567 | 0.703 | 38.4 |
> | **A-Mem** |Qwen2.5-7B | **0.730** | **0.961** | 18.8 |
>
> **Results on WebShop task with GPT-4o as foundation model.**
>
> | Method  |Final Reward | Token ($*10^3$) |
> |---|---|---|
> | LoCoMo (Truncate) |13.82  | **0.99** |
> | LoCoMo |  **25.48**  | 5.30 |
> | **A-Mem**  | 24.50 |  1.84 |
>
> Reference:
>
> [1] Zhong, Wanjun, et al. "Memorybank: Enhancing large language models with long-term memory." AAAI 2024.
>
> [2] Liu, Nelson F., et al. "Lost in the middle: How language models use long contexts." arXiv:2307.03172 (2023).
>
> [3] Kaddour, Jean, et al. "Challenges and applications of large language models." arXiv:2307.10169 (2023).
>
> [4] Yang, Zhilin, et al. "HotpotQA: A dataset for diverse, explainable multi-hop question answering." arXiv:1809.09600.
>
> [5] Kwiatkowski, Tom, et al. "Natural questions: a benchmark for question answering research." TACL (2019).
>
> [6] Maharana, Adyasha, et al. "Evaluating very long-term conversational memory of llm agents." arXiv:2402.17753 (2024).
>
> [7] Yao, Shunyu, et al. "Webshop: Towards scalable real-world web interaction with grounded language agents." NIPS 2022.

---

> > ### Comment · Reviewer_3BrE · 2025-08-05
> >
> > Thanks for your response, I have no further questions.

---

> > > ### Author Response · Authors · 2025-08-05
> > >
> > > We're grateful that our response helped clarify your concerns. Thank you again for your thoughtful feedback.

---

### Official Review · Reviewer_qBs5 · 2025-06-22

**Clarity:** 3
**Significance:** 2
**Originality:** 3
**Rating:** 4
**Confidence:** 4

**Summary:**

This paper proposes a novel memory construction method for LLM-based systems, termed A-Mem. Unlike traditional approaches such as RAG, A-Mem offers greater flexibility through dynamic memory organization. New memories can be added, linked to similar historical memories, and used to trigger updates to the contextual representations and attributes of existing ones, enabling the memory network to evolve continuously. An LLM serves as a judge to determine memory similarity and guide the refinement of the memory network. Empirical experiments conducted on six foundation models demonstrate significant improvements over existing state-of-the-art baselines.

**Questions:**

As noted in the "Weaknesses" section, if we already have a sufficiently capable LLM judge with comprehensive prior knowledge, is it still necessary to maintain a separate memory network? Have you conducted any experiments to evaluate the necessity of maintaining such a network under the assumption that a capable LLM judge is already available?

**Ethical Concerns:**

["NO or VERY MINOR ethics concerns only"]

**Final Justification:**

The authors' responses have largely addressed my concerns. Therefore, I decided to raise my score to 4.

**Limitations:**

yes

**Quality:**

2

**Strengths And Weaknesses:**

Strengths:
1. The paper addresses a significant and timely problem—how to construct memory systems for LLMs—which is a critical research direction in advancing long-term reasoning and context management.
2. The concept of a dynamically evolving memory structure is intuitive and compelling, as it mirrors the way human memory functions.
3. The paper not only proposes this idea but also provides an implementation and empirical validation through a series of experiments.

Weaknesses:
As acknowledged in the "Limitations" section, the efficiency and overall performance of A-Mem heavily rely on the capabilities of the LLM judge, which is responsible for determining memory similarity and guiding the refinement of the memory network. Conceptually, if we already possess a sufficiently capable LLM judge with comprehensive prior knowledge, one might question the necessity of maintaining a separate memory network at all.

---

> ### Author Rebuttal · Authors · 2025-07-31
>
> Thank you for your review. We sincerely appreciate your recognition that A-Mem addresses a significant and timely problem in LLM memory construction, which you identified as critical for advancing long-term reasoning. We're grateful you found our dynamically evolving memory structure intuitive and compelling, noting how it mirrors human memory functions. Thank you also for acknowledging our comprehensive approach that provides concrete implementation and empirical validation across six foundation models, demonstrating significant improvements over existing baselines.
>
> ***
> > ***Q1. If we already have a sufficiently capable LLM judge with comprehensive prior knowledge, is it still necessary to maintain a separate memory network? Have you conducted any experiments to evaluate the necessity of maintaining such a network under the assumption that a capable LLM judge is already available?***
>
> Even if we have a sufficiently capable LLM judge with comprehensive prior knowledge, memory systems remain essential for LLM agents. We provide a detailed explanation from the following perspectives.
>
> **Necessity of memory system for LLM.** Large language models suffer from fundamental architectural limitations including finite context windows, quadratic attention complexity, and the "lost in the middle" problem where performance degrades significantly when accessing information in the middle of long contexts, even for explicitly long-context models [1-2]. Given these existing architectural constraints, it is impractical to feed all past interaction history of agents into LLMs while maintaining optimal performance. Therefore, memory systems enable LLM agents to transcend context window limitations by persistently storing experiences, abstracting knowledge, and supporting self-evolving capabilities for long-term agent-environment interactions [3].
>
> **LLMs in our A-mem.**  In our agentic memory system, we utilize LLMs to organize memory in a Zettelkasten-style storage structure. Additionally, we design a retriever to identify and retrieve similar information when accessing memory. By leveraging both LLMs and retrievers, our A-Mem provides condensed, informative context that enables LLM agents to transcend context window limitations with reasonable token consumption.
>
> **Experiment validation.** In the Table 1-2 of our initial paper version, we conducted experiments comparing our approach against the LoCoMo method, which relies on LLMs to process all context information without memory retrieval. Despite LoCoMo [4] using approximately 7x more tokens than our A-Mem, it still underperforms compared to our method.
>
> **Conclusion.** Due to the inherent limitations of LLM architecture, models cannot process excessively long contexts while maintaining optimal performance. With our A-Mem design, LLM agents can achieve superior performance through efficient memory retrieval mechanisms.
>
>
> References:
>
> [1] Liu, Nelson F., et al. "Lost in the middle: How language models use long contexts." arXiv preprint arXiv:2307.03172 (2023).
>
> [2] Kaddour, Jean, et al. "Challenges and applications of large language models." arXiv preprint arXiv:2307.10169 (2023).
>
> [3] Zhang, Zeyu, et al. "A survey on the memory mechanism of large language model based agents." ACM Transactions on Information Systems (2024).
>
> [4] Maharana, Adyasha, et al. "Evaluating very long-term conversational memory of llm agents." arXiv preprint arXiv:2402.17753 (2024).

---

> > ### Comment · Reviewer_qBs5 · 2025-08-04
> >
> > Thank you for responses, which have largely addressed my concerns. Therefore, I have decided to raise my score to 4.

---

> > > ### Author Response · Authors · 2025-08-04
> > >
> > > Thank you for your time and for reconsidering our work after our rebuttal. We are very glad to hear that our responses helped address your concerns. We sincerely appreciate your constructive feedback throughout this process.

---

### Official Review · Reviewer_zYJp · 2025-07-02

**Clarity:** 2
**Significance:** 3
**Originality:** 3
**Rating:** 5
**Confidence:** 4

**Summary:**

This paper present agentic memory, A-MEM, inspired by the Zettelkasten method. The agentic memory is composed of:
1) note construction: record original interaction, keywords, tags, description, etc.
2) link generation: use cosine similarity to search for the k most relevant memories, then an LLM is prompted to analyse potential common attributions.
3) memory evolution: relevant memories are evolved when a new memory is added. Allowing the system to learn as it accumulates experience.
Finally, the paper performs extensive studies, ablation and hyper-parameter sensitivity.

**Questions:**

* LoCoMo is both a baseline and a dataset?
* Why is the memory usage the exact same for every method in Table 4? The column might not be necessary since memory usage is a function of memory size.
* Does average token length in Table 1 includes all the tokens used to generate the meta data during note writing and memory evolution?

**Ethical Concerns:**

["NO or VERY MINOR ethics concerns only"]

**Final Justification:**

The proposed agentic memory system is novel, carefully evaluated and timely. The method also consistently improves over the baseline without being more expensive.

**Limitations:**

yes.

**Quality:**

3

**Strengths And Weaknesses:**

# Strengths

## Novelty
* An agentic perspective on memory system is timely and novel.

## Evaluation
* The datasets LoCoMo and DialSim look well designed to test memory systems.
* Good ablation studies.
* A-mem provides consistent improvement over the baselines.

# Weaknesses

## Expensive
* The proposed approach sounds expensive in terms of 1) memory since it registers the original interaction content, and 2) LLM calls since the LLM must generate tags and contextual description of each note. Furthermore, the LLM must analyze potential connections between notes based on their potential common attributes.

## Presentation
* I believe the presentation could be improved. For example, in Figure 2 several terms appear twice: box n+1 and n+2, and memory evolution and evolve.
* In section 4.4, please include links to Table 3 in the text.
* Add average number of tokens to Table 3.

---

> ### Author Rebuttal · Authors · 2025-07-31
>
> Thank you for your positive evaluation of our work. We appreciate your recognition of our key contributions regarding A-MEM, the effectiveness of our Zettelkasten-inspired approach with its three core components, and the comprehensive evaluation demonstrating consistent improvements over baselines. We are grateful for your acknowledgment of the well-designed LoCoMo and DialSim datasets, the thoroughness of our ablation studies, and the novelty of our approach in enabling systems to learn and evolve through accumulated experience. We address the points you raised to further enhance the clarity of our work.
>
> ***
> > ***Q1. Whether the approach is expensive because of LLM will be used multiple times for each memory?***
>
> As reported in the cost-efficiency analysis in the initial version of the paper, we demonstrate that memory operations using GPT-4o-mini cost less than $0.0003. Here, we provide a detailed analysis of the cost for memory operations. It should be noted that all price calculations are based on OpenAI pricing information from five months ago, and prices may fluctuate over time.
>
> The analysis of the A-MEM memory system reveals the following key metrics:
>
> - System Prompt Overhead: 363 tokens per operation
> - Average Input Processing: 1,042.00 tokens per memory operation
> - Average Output Generation: 131.00 tokens per memory operation
> - Total Tokens Per Operation: 1,173.00 tokens
> - For per memory storage, using the gpt-4o-mini api costs an average of 5.39 seconds, and using locally-hosted llama3.2 1B model costs an average of 1.12 seconds with a single GPU.
>
> Based on the GPT-4o-mini pricing model (`$`0.150/1M tokens for input; `$`0.600/1M tokens for output):
>
> - **Input Cost Calculation:**
> 1,042.00 tokens × (`$`0.150/1M tokens) = `$`0.0001563 per operation
> - **Output Cost Calculation:**
> 131.00 tokens × (`$`0.600/1M tokens) = `$`0.0000786 per operation
> - **Total Cost Per Memory Operation:**
> `$`0.0001563 + `$`0.0000786 = `$`0.0002349
> - **Projected Cost at Scale:**
> For 1 million memory operations:`$`0.0002349 × 1,000,000 = `$`234.90
>
> **Conclusion.** The A-MEM system demonstrates reasonable resource efficiency at approximately `$`0.0002349 per memory operation (`$`234.90 per million operations). Despite requiring multiple LLM calls during memory processing, this cost structure remains economically viable and scales predictably.
>
>
> ***
> > ***Q2. Could you please revise Figure 2 to remove duplicate terms like "box n+1" and "evolve"?***
>
> In the initial version of our paper, we used the terms "box n+1" and "box n+2" to indicate that these two boxes are newly added in the link generation process. In the final version, we will revise the figure for better clarity.
>
> ***
> > ***Q3. In section 4.4, please include links to Table 3 in the text.***
>
> Thank you for the suggestion. We will revise to include hyperlinks to Table 3 in Section 4.4 text.
>
> ***
> > ***Q4. Add the average number of tokens to Table 3.***
>
> For the "w/o LG & ME" method, the average token count is 605. For the "w/o ME" method, the average token count is 2,371. We will include this information in the final version of our paper.
> Note that the average token count is computed based on the QA test phase, excluding the memory construction phase.
>
> ***
> > ***Q5. Can you clarify the role of LoCoMo, as it appears to be used as both a baseline and a dataset?***
>
> LoCoMo [1] provides a dataset and a basic memory evaluation method in their paper. In the initial version of our paper, we used the same abbreviation to represent our method. In the final version, we will revise our method's abbreviation to avoid confusion and improve clarity.
>
> ***
> > ***Q6. Why is the memory usage the exact same for every method in Table 4?***
>
> For fair comparison, we use the same text embedding model across all three memory systems, resulting in the embedding storage occupying the majority of memory usage. We will update Table 4 in the final version of the paper to reflect this clarification.
>
> ***
> > ***Q7. Does the average token length in Table 1 include all tokens used for metadata generation during both note writing and memory evolution?***
>
> Since memory construction and QA evaluation are separate processes in the datasets, we do not include the average token cost of memory construction in Table 1. For the memory construction token cost, please refer to the response to Q1.
>
> Reference:
>
> [1] Maharana, Adyasha, et al. "Evaluating very long-term conversational memory of llm agents." arXiv preprint arXiv:2402.17753 (2024).

---

> > ### Author Response · Authors · 2025-08-06
> > **Kindly Reminder**
> >
> > Dear Reviewer,
> >
> > We have submitted our detailed rebuttal addressing all the raised concerns. We would greatly appreciate it if you could review our response and let us know if any aspects require further clarification.
> >
> > Best regards,
> >
> > Authors

---

### Author Response · Authors · 2025-08-09
**General Response by Authors**

Dear Area Chairs and Reviewers,

We appreciate the reviewers' time, valuable comments, and constructive suggestions. The rebuttal process was highly productive, allowing us to clarify key aspects of our work, provide additional analyses, and run new experiments. We are grateful that our responses were well-received, leading to two reviewers explicitly raising their scores and all others expressing satisfaction.

---

### **Strengths acknowledged by the reviewers:**

1.  **Novelty and Originality:** The A-MEM method was consistently praised as a novel, innovative, and significant memory paradigm that supports unique evolutionary capabilities, departing from traditional fixed-schema approaches (Reviewers **zYJp**, **qBs5**, **3BrE**, **rgsn**).
2.  **Dynamic & Adaptive Structure:** The Zettelkasten-inspired design, featuring dynamic memory organization, automatic semantic linking, and autonomous evolution, was highlighted as intuitive, compelling, and highly adaptable (Reviewers **qBs5**, **3BrE**, **rgsn**).
3.  **Strong Performance & Efficiency:** The method demonstrated consistent improvement over baselines in long-term conversational tasks while maintaining low resource consumption (Reviewers **zYJp**, **3BrE**).
4.  **Thorough Evaluation:** The paper was recognized for its comprehensive experiments, including multiple baselines, diverse metrics, well-designed datasets, and strong ablation studies (Reviewers **zYJp**, **rgsn**).

---

### **Main concerns raised by reviewers:**

1.  **Computational Overhead and Scalability:** Concerns about the cost of multiple LLM calls and a lack of convincing analysis on how retrieval time and storage scale as memories accumulate (Reviewers **zYJp**, **3BrE**, **rgsn**).
2.  **Evaluation Limitations:** The evaluation was seen as limited by its focus on QA datasets without demonstrating applicability in broader agentic systems, and some experiments were missing comparisons against all baselines (Reviewers **3BrE**, **rgsn**).
3.  **Conceptual and Design Questions:** Questions were raised about the fundamental need for a memory system given powerful LLMs, the reasonableness of similarity-based linking, and the simplicity of using a single encoder for heterogeneous inputs (Reviewers **qBs5**, **3BrE**, **rgsn**).

All these concerns have been successfully addressed during the rebuttal phase with comprehensive responses and substantial new experiments, resulting in positive feedback and increased scores.

---

### **Revisions and Clarifications in the Rebuttal Phase:**

1.  **New Agent System Experiments:** To demonstrate broader applicability, we conducted **new experiments on agentic tasks**, evaluating A-Mem on QA search (HotpotQA, NQ) and web shopping (WebShop). Results showed A-Mem achieves comparable or superior performance to baselines while using only **33-50% of the token budget**, validating its effectiveness and efficiency in practical agent scenarios.
2.  **Detailed Cost & Scalability Analysis:** We provided a granular breakdown of computational costs, showing operations are highly efficient (avg. **$0.0002349 per operation** with GPT-4o-mini). We also clarified our scaling analysis, explaining how retrieval time is dominated by corpus size ($N$) and remains largely independent of the number of retrieved items ($k$).
3.  **Expanded Baseline Comparisons & Ablation Study:** We completed the evaluation on the **DialSim dataset by running all missing baselines**, confirming A-Mem's state-of-the-art performance across all metrics. We also conducted a **new ablation study** on our memory vector representation, empirically proving that each of its four components is critical for performance.
4.  **Conceptual Clarifications & Case Studies:** We provided detailed arguments for the necessity of external memory systems due to core LLM limitations. We also presented **concrete case studies** to illustrate how our two-stage linking mechanism captures nuanced causal and thematic relationships that transcend simple vector similarity.

---

We sincerely appreciate your valuable time and consideration!

Thanks and regards,

Authors

---

### Decision · Program_Chairs · 2025-09-17

**Decision:**

Accept (poster)

**Comment:**

This paper presents A-MEM for agentic memory inspired by Zettelkasten. It consists of note construction, link generation, and memory evolution. Reviewers noted the originality of the contribution and the importance of the problem, as well as the quality of the experiments, which are comprehensive (4 baselines, multiple tasks measured via both F1 and BLEU scores + ablations, scaling, memory). After discussion, all reviewers support accepting the paper.